# Evaluating low cost topographic surveys for computations of conveyance.

Hubert T. Samboko[1], Sten Schurer[1], Hubert H.G. Savenije[1], Hodson Makurira[2], Kawawa Banda[3], Hessel Winsemius[1, 4, 5]

[1]Department of Water Resources, Faculty of Civil Engineering and Geosciences, Delft University of Technology, Stevinweg 1, 2628 CN, Delft, Netherlands
[2]Department of Construction and Civil Engineering, University of Zimbabwe, Box MP 167, Mt. Pleasant, Harare, Zimbabwe
[3]Department of Geology, Integrated Water Resources Management Centre, University of Zambia, Great East Road Campus, P.O. Box 32379, Lusaka, Zambia
[4]Deltares, Delft, the Netherlands
[5]Rainbow Sensing, The Hague, the Netherlands

*Correspondence to*: Hubert T. Samboko (hsamboko@gmail.com)

**Abstract.** Rapid modern technological advancements have led to significant improvements in river monitoring using Unmanned Aerial vehicles (UAVs), photogrammetric reconstruction software and low-cost Real Time Kinematic Global

Navigation Satellite System (RTK GNSS) equipment. UAVs allow for the collection of dry bathymetric data in environments that are difficult to access. Low-cost RTK GNSS equipment facilitate accurate measurement of wet bathymetry when combined with subaqueous measuring tools such as Acoustic Doppler Current Profilers (ADCPs). Hydraulic models may be constructed from these data, which in turn can be used for various applications such as water management, forecasting, early warning and disaster preparedness by responsible water authorities, and construction of river rating curves. We hypothesize

that the reconstruction of dry terrain with UAV-based photogrammetry combined with RTK GNSS equipment leads to accurate geometries particularly fit for hydraulic understanding and simulation models. This study sought to (1) compare open source and commercial photogrammetry packages to verify if water authorities with low resource availability have the option to utilise open source packages without significant compromise on accuracy; (2) assess the impact of variations in the number of Ground Control Points (GCPs) and the distribution of the GCP markers on the quality of Digital Elevation Models (DEMs), with a

particular emphasis on characteristics that impact hydraulics; and (3) investigate the impact of using reconstructions based on different GCP numbers on conveyance and hydraulic slope. A novel method which makes use of a simple RTK tie line along the water edge measured using a low cost but highly accurate GNSS is presented so as to correct the unwanted effect of lens distortion ('doming effect') and enable the concatenation of geometric data from different sources. Furthermore, we describe how merging of the dry and wet bathymetry can be achieved through gridding based on linear interpolation. We tested our

approach over a section of the Luangwa River in Zambia. Results indicate that the open-source software photogrammetry package is capable of producing results that are comparable to commercially available options. We determined that GCPs are essential for vertical accuracy, but also that an increase in the number of GCPs above a limited amount of 5 only moderately increases the accuracy of results, provided the GCPs are well spaced both in horizontal and vertical dimension. Furthermore,

insignificant differences in hydraulic geometries among the various cross sections are observed, corroborating the fact that a limited well-spaced set of GCPs is enough to establish a hydraulically sound reconstruction. However, it appeared necessary to make an additional observation of the hydraulic slope. A slope derived merely from the UAV survey was shown to be prone to considerable errors caused by lens distortion. Combination of the photogrammetry results with the RTK GNSS tie line was shown to be essential to correct the slope and made the reconstruction suitable for hydraulic model setup.

**Key Words:** Unmanned Aerial Vehicle, Digital Elevation Model, Ground Control Point, Conveyance

# 1    Introduction

Traditionally, flow measurements are performed through the use of current meters. A combination of measured depth and velocities across a profile can be integrated to calculate the total discharge. In order to attain continuous discharge data, river stage is recorded and plotted against corresponding discharge measurements to produce rating curves (Herschy, 2009; Mosley and McKerchar, 1993). Ideally, discharge measurements are carried out over a wide range of river stages. The low and medium river stages are usually relatively easy to record whereas the high river stages are difficult as they are associated with dangerous conditions such as floods and inaccessible terrains. Peaks are also easy to miss, as deployment of personnel and materials takes time. Due to these difficulties, high stage discharge measurement is usually extrapolated from the rating curve. On the other hand, there is the risk of high variability in low flow measurements as a result of changing bed configurations, particularly in sand rivers which change every season. Measurement are usually taken at one particular point frequently despite physical changes in the profile. These problems lead to high levels of uncertainty in discharge estimates which makes it difficult for water authorities to understand runoff generation processes especially during high flows when management is mostly required (Petersen-Øverleir et al., 2009). Another limitation is the time validity of the measurements which strongly depends on factors such as river bed degradation, river course changes after floods and overspill or ponding in areas adjoining the stream channel (Herschy, 2009; Rantz and Others, 1982). Changes in the geometry of the river due to these factors affect the rating curve output. Therefore, measurements may cease to be valid across time.

Using a hydraulic modelling strategy has become an alternative for discharge estimation (Mansanarez et al., 2019). Physically based river rating is based on capturing geometry in a power law expression. The physically based river rating makes use of the fact that river flow is a function of river slope, river-bed roughness and channel geometry. In this instance discharge calculations of flow require information about the geometry of the channel in question (Costa et al., 2000). One of the most commonly used equations is Manning's formula which is based on steady and uniform flow (Chow, 1959).

The Manning equation can be rewritten as the power law function Eq. (1):

$$Q = n^{-1}\sqrt{i}(AR^{2/3}),$$

(1)

where $Q$ is discharge [m³/s], $n$ the Manning's roughness coefficient, $i$ is the bottom slope [-], $A$ is the cross-sectional area [m²] and $R$ the hydraulic radius [m], [s m$^{-1/3}$].In this equation; the first part $(n^{-1}\sqrt{i})$ depends on the bottom slope and channel roughness, the second part $(AR^{2/3})$, depends on the cross-sectional geometry. We refer to $A$ and $R$ collectively as "hydraulic geometry" and $AR^{2/3}$ as the "conveyance".

The hydraulic geometry is a critical input in the production of rating curves (Zheng et al., 2018). Improvements in technology have allowed for a wide range of options for the establishment of geometry. These methods include survey equipment (levels, theodolites, Differential GNSS), Ground Penetrating Radar, sensors mounted on satellites, aeroplanes, kites, unmanned aerial vehicles (UAV), hot air balloons (Feurer et al., 2008; Salamí et al., 2014). In general, manned aircraft which carry cameras are much more costly than other forms of image data collection (Yang et al., 2006). A low-cost means of collecting geometry is through systematic capturing of images from one or multiple cameras mounted on an unmanned aerial vehicle (UAV). Advancements in technologies have resulted in the ability of surveyors to collect very high-resolution geometrical data in difficult to access places (Samboko et al., 2019).

The advantages of using UAVs are, (i) the portability of UAVs; ii) the option to self-design and modify integrated sensors; (iii) the availability of open source and user-friendly data processing software; (iv) the collection of data in difficult to access terrains and; (v) the relatively low-cost of basic UAVs (Gindraux et al., 2017). UAVs, which operate at low altitudes, have a much higher spatial resolution than satellites and are not limited in temporal resolution. When used in combination with Ground Control Points (GCPs), UAVs are capable of reconstructing dense and accurate terrains. Satellites with high spatial resolution usually have long revisit intervals. Only a very limited amount of studies so far have used UAVs to collect data for hydraulic model purposes.

The application of UAV based imagery for dry bathymetry reconstruction is relatively well practiced and documented (Coveney and Roberts, 2017; Gustafsson and Zuna, 2017; Yao et al., 2019). Unfortunately, most low-cost UAVs with RGB sensors are incapable of mapping the geometry under water. Given that many large rivers of interest are perennial, the common practice is to use subaqueous measuring tools such as Acoustic Doppler Current Profilers (ADCPs) to determine the 'wet' bathymetry of rivers (Vermeyen, 2007; Zedel et al., 2018). Depth profiling has become more affordable with recently developed low-cost echo sounding devices, which are a viable alternative for typically high cost ADCP devices. This was recently shown by Broere et al. (2021) who used a low-cost echo sounder to detect macro-plastics in streams. However, most

ADCPs or echo sounders are equipped with consumer grade GNSS instruments with 2 meter accuracy. This level of accuracy is unacceptable for accurate hydraulic modelling purposes.

The demand for both accurate and accessible measurements have driven the development of low-cost GNSS instruments (Glabsch et al., 2009; Poluzzi et al., 2019). Recent multi frequency GNSS receivers are affordable, lightweight and are able to function in static and dynamic mode. They also act as accurate replacements for the on-board consumer grade GNSS instruments as they have been proven to be highly accurate and applicable as substitutes for traditional methods (Cina and Piras, 2014). A low cost GNSS chipset (ZED-F9P) was released by U-blox in 2019. In this study we use this chipset on a

breakout board of Ardusimple, type SimpleRTK2B. The set is uniquely capable of receiving corrections from both the L1 and L2 bands (u-blox, 2021). Research conducted using the SimpleRTK2B GNSS set have confirmed its ability to produce results comparable to accurate geodetic measurements (Hamza et al., 2020, 2021).

Apart from the impact of instrumental (GNSS, ADCP and UAV) inaccuracies on hydraulic geometry, there are more factors to consider for conveyance calculations. These factors can be divided into three groups; (i) pre-flight (UAV, flight application,

flight path and site selection), (ii) flight settings (camera angle, direction, velocity, altitude, light intensity, wind speed, overlap) and (iii) post-flight processing (photogrammetry software, camera lens distortion, GCP configuration and slope). There have been a number successful attempts to review and evaluate best practices for pre-flight and flight settings of UAV acquisition systems, orientation and regulation (Abou Chakra et al., 2020; Chaudhry et al., 2020; Seifert et al., 2019; Yao et al., 2019). We proceed by evaluating the four constituents (photogrammetry software, GCP configuration, camera lens distortion and

slope) of post-flight processing which are important for accurate reconstruction of hydraulic geometry.

Firstly, the post-flight processing of UAV derived imagery is largely and increasingly facilitated by 'structure-from-motion' (SfM) photogrammetry software. It offers image processing workflows which are easier to work with than traditional photogrammetry techniques. SfM based approaches have been successfully used in various applications such as soil and coastal erosion and lava emplacement (Castillo et al., 2012; James and Robson, 2012; James and Varley, 2012; Smith et al., 2015).

Unfortunately, SfM photogrammetry requires software which is usually available at a cost beyond the reach of most researchers and other interested parties. Some of the more common software packages are (commercial) Pix4D, Agisoft meta-soft and (non-commercial and open-source) OpenDroneMap (ODM). Several researchers have made some comparisons between the commercially available software (Alidoost and Arefi, 2017; Grussenmeyer and Khalil, 2008; Probst et al., 2018). ODM is an open-source software which can be used to generate digital elevation models and other photogrammetry results. Not only does

the non-commercial nature of ODM make it more accessible to researchers and practitioners with limited resources, it also presents an opportunity to tweak and investigate the impact of individual variables on the output (Burdziakowski, 2017).

The second aspect of post-processing which is important for hydraulic geometry is the GCP configuration. Similar to ADCPs, UAVs are equipped with a consumer grade GNSS with an accuracy of 2 meters. This means that all UAV based images and

outputs of photogrammetry have a maximum error of 2 meters (Udin and Ahmad, 2014). For the purposes of hydraulic modelling, this inaccuracy is unacceptable, therefore, the application of GCPs is paramount. A number of studies have investigated the number and distribution of GCPs necessary to generate accurate elevation models (Awasthi et al., 2019; Bandini et al., 2020; Ferrer-González et al., 2020; Rock et al., 2011). However, studies have not gone as far as to investigate how to adjust the number and distribution of GCPs specifically for the purposes of modelling flow in hydrodynamic conditions. For instance, the specific impact on hydraulic geometry of GCP proximity to a flowing river is largely unknown. This particular information would be handy for water managers who aim to survey the dry and wet bathymetry of a river using low-cost technologies.

The third aspect of post-processing which is important for hydraulic geometry is camera lens distortion. Investigation into camera lens distortion can be traced as far back as 1919 when A. Conrady developed the decentering distortion method (Conrady, 1919). Based on the decentering model, Brown developed the Brown-Conrady model (Brown, 1971; Clarke and Fryer, 1998). There have been a number of improvements and modifications to the Brown-Conrady model with respect to different applications (Beauchemin and Bajcsy, 2001; Ma et al., 2003; Shah and Aggarwal, 1996). Despite tremendous improvement in terms of reduced distortion, some DEMs show systematic broad scale deformation which is known as the 'doming effect' (also known as the 'bowling effect') (Javernick et al., 2014; Rosnell and Honkavaara, 2012). The doming effect emanates from inaccuracies in modelling the radial distortion of camera lens (Fryer, John & Mitchell, 1987). This fundamental drawback makes it difficult to fully exploit the potential of SfM products in many situations such as gradient sensitive applications, e.g. rainfall runoff and slope estimation. Some guidelines for avoiding the doming effect have been outlined (James and Robson, 2014a). A novel method which aimed at correcting the doming effect was presented by Magri (2017), who iteratively applied a planarity constraint through a Bundle adjustment framework. The results were encouraging as they concluded that it was possible to mitigate the doming effect through manipulation of the bundle adjustment process. Bundle adjustment is a technique for calculating the errors that occur when we transform the XYZ location of a point in the environment to a pixel point on a camera image.

Documentation from ODM suggests that making use of a configuration called Fixed Camera Parameter (FCP) can help reduce the doming effect (ODM, 2021). The FCP turns off camera optimisation while performing bundle adjustment. This is because in certain circumstances, particularly when mapping linear (low amplitude, limited features) topographies, bundle adjustment performs poor estimation of distortion parameters (Griffiths and Burningham, 2019).

Finally, in order to estimate flow based on the Manning's formula (Equation 1), it is important to accurately measure the slope of the terrain. Similar to hydraulic geometry, there is growing interest in non-contact methods of estimating slope. Common methods of slope measurement require accurate point data measured using GNSS and geodetic based methods. It is possible to extract elevations from photogrammetry outputs and derive slope, however, the accuracy of this method is largely unknown.

Ultimately, the factors (photogrammetry software, GCP configuration, lens distortion and slope) which affect hydraulic geometry can be evaluated in terms of their impact on discharge or flow proxies such as conveyance. A study was conducted by Mazzoleni (2020) on the potential for using UAV derived topography for hydraulic modelling. The study concluded that these topographies extracted from UAVs presented results comparable to LIDAR and RTK GNSS-based topographies. However, it did not accurately measure the permanently wetted bathymetry of the river. Rather, the study mechanically filtered

out the river which brought about some uncertainty. A similar study which investigated the impact of the number of GCPs on flood risk model performance concluded that UAVs could successfully be used for data collection as long as a minimum number of control points were utilised (Coveney and Roberts, 2017). Nevertheless, the study was located in a large city and thus, did not include the measurement of inundated areas, nor did it focus on the ability to reconstruct typical hydraulic properties.

The practical utility of accurate hydraulic geometry for flow estimation is unquestionable (Gleason, 2015). However, there exists minimal research on how the factors which affect the accuracy of geometry can be adjusted to improve the quality of elevation models in hydrodynamic environments and when applied for the ultimate purposes of discharge estimation. Furthermore, earlier contributions have not put the focus on the ability to reproduce hydraulic geometry characteristics and have not focused on the entire bathymetry (including the permanently wet river bed section). Hence, this paper investigates if

low-cost methods for data collection and processing, i.e. a combination of precise wet bathymetry points with UAV photogrammetry, can be used to provide satisfactory quality elevation models for hydraulic models, quantified in hydraulic geometry characteristics. In this paper, a novel and practical method of correcting the doming effect using data collected using a low cost GNSS, mounted on a mobile cart is applied. We tested the methods on the Luangwa River in Zambia.

    This paper is organised as follows: section 2 describes the methodology and gives a brief outline of what materials were used

in the study. In Section 2.1 describe the study area (Luangwa Basin). Furthermore, the methodology section outlines how flow estimation was determined and software packages were compared. Furthermore, Section 3 presents results and a discussion of the results. We conclude with section 4 which presents a conclusion and recommendation for future studies.

    We investigate the following research questions and determine whether the said factors have a significant effect on the accuracy of results. These are:

1. Can the freely available (Open Source) ODM software package produce results that are comparable to commercial packages such as Agisoft Metashape?

    2. What is the optimal GCP number and GCP distribution necessary to reconstruct accurate elevation models?

    3. What impact do elevation models, reconstructed based on different GCP numbers have on hydraulically simulated conveyance and hydraulic slope?

## 2 Materials and Methods

This section first describes the data collection procedures, including flight plan, collection of ground control points, dry and wet bathymetry. Then it describes which experiments are conducted to investigate our research questions.

### 2.1 Study site

The study was conducted along the Luangwa River, South of the Luangwa Bridge. The Basin has a catchment area of approximately 160,000 km$^2$. The Luangwa River originates in the Mafinga Hills in the North-Eastern part of Zambia and is approximately 850 km in length, flowing in South-Western direction. The river drains into the Zambezi River, shaping a broad valley along its course. The river has naturally created a valley, which is well-known for its abundant wildlife and relatively pristine surroundings (WARMA, 2016). The study area is shown on *Figure 1*.


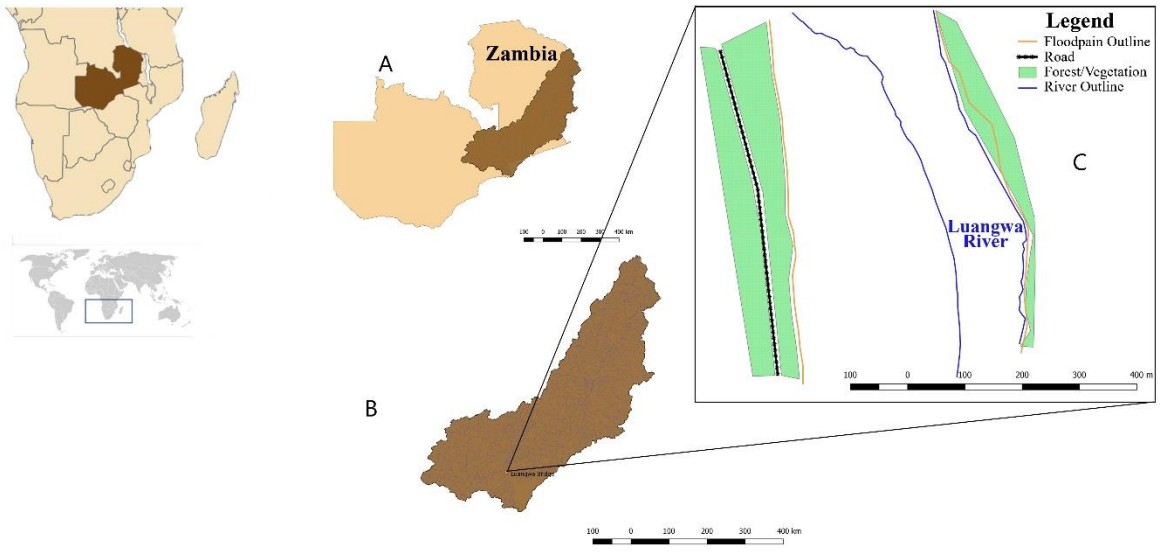

**Figure 1 Study area map A: Zambia, B: Luangwa River, C: Mapped Area**

The data collection was conducted in the late stages of the dry season (December, 2019) to maximise the visible floodplain.

To optimise on access and data collection efficiency, the reach which was chosen is relatively straight with low sinuosity. The
wet river channel is however meandering within the floodplain. The channel is also wandering and braiding, especially during low flows. The longitudinal section has a gentle gradient (approximately, 1:5000), which is difficult to identify without the

use of accurate survey instruments. At high water, the river cross section is approximately 400 meters in width, with a maximum depth of approximately 8 meters. At the time of data collection (in the dry season) the flowing water had a maximum depth of 2 meters. The channel substrates are alluvial, comprised of sand. Erosion, siltation and sedimentation are therefore highly prevalent occurrences. It is not unusual to see the river channel in a different location after every wet season or after a heavy storm event due to high morphological activity.

## 2.2    Data acquisition

The data acquisition basically consists of two parts, data collection and data-processing. The data collection includes measuring ground control points, measuring the river bathymetry and collection of UAV images.

### 2.2.1    Low cost GNSS equipment

In 2019, U-blox launched the ZED-F9P chip capable of receiving satellite signals in the lower and upper bands (L1 and L2) from the BeiDou, Galileo, GLONASS and GPS constellations. The ZED-F9P chip was integrated with an Arduino simpleRTK2B board which can function in RTK mode, produced by Ardusimple. The board can transmit or receive Radio Technical Commission for Maritime (RTCM) corrections and can be configured by the user using u-center, a freely available open source software (u-blox, 2021). The simpleRTK2B set is low-cost (receiver 172 Euro and patch antenna 50 Euros at the time of writing) with the possibility to acquire <1cm level precision with base-rover and <1cm level precision with RTCM corrections. The exact accuracy depends on multiple factors including the used antenna, the satellite reception quality and amount, the accuracy of the base station surveyed location, and the baseline distance. Long-Range radio antennas were used to communicate RTCM messages. *Figure 2 (a)* shows the SimpleRTK2B Base and Rover which was used to measure marker points. *Figure 2 (b)* shows the simpleRTK2B setup on site. At the initial stages of configuration, the board was connected to a laptop which also provided power supply and data storage through a USB port. Upon realisation that a laptop would not be able to supply power for a prolonged period of time in harsh fieldwork conditions, we replaced it with two 20 000mAh power banks and a Raspberry Pi. The time between starting the base station and actually beginning to take measurements using the rover has an impact on accuracy i.e. an extended time period results in better results because the base is able to survey its location more precisely over time.

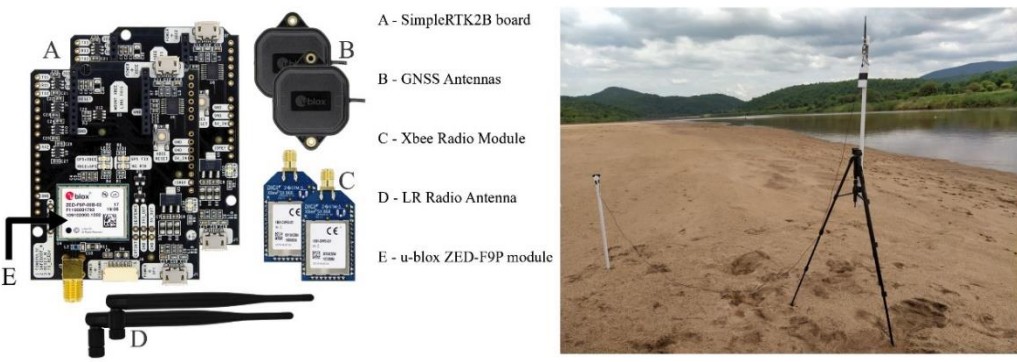

**Figure 2 (a) RTK GNSS Set (b) RTK GNSS Base Station Setup along the Luangwa River Floodplain**

### 2.2.2    Flight Plan

GCPs were recorded using RTK GNSS equipment on a 1 km long floodplain. Flights were conducted at two different heights (90m and 100m) at a constant speed of 10 m/s, a $10^0$ camera angle used to optimise on 3D reconstruction results. The two flight patterns were separated by 20 degrees from each other so as to limit the effects of image lens distortion. The side and forward image overlap was set to 80%. Figure 3 shows the flight paths of the two patterns which were flown. The UAV used is a DJI Phantom 4 Advance with a 12 Megapixel FC330 RGB camera with a focal length of 3.61 mm. A flight planning android application called Pix4D Capture was used to control the autonomous flights. This application was chosen due to its

capability to tilt the camera forward during the image capturing process, important to capture more depth information than when using a nadir-looking configuration. The coordinate system was set to WGS 84 / UTM zone 36S (EPSG::32736).

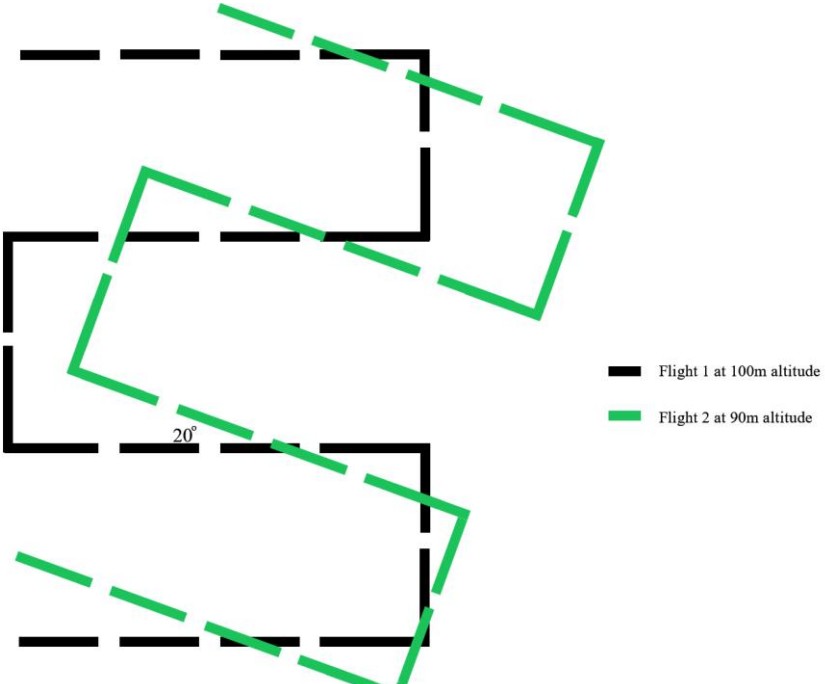

**Figure 3 Flight paths flown at two different heights (90m and 100m) at a 20 degree angle to each other.**

### 2.2.3    Dry river bathymetry

In order to refine the camera calibration parameters and to optimise the geometry of the output, GCPs have to be used. The dry bathymetry data collection can be divided into two procedures; placing the GCPs on the ground and collecting the images. A total of 17 GCP markers were placed on the floodplain, with some being closer to the road, others more in the middle of the dry floodplain and the last closer to the water line. *Figure 4* shows the location of the GCPs in relation to the floodplain. The GCPs were placed on one side of the floodplain because the other side was steep and covered with dense vegetation. An effort was made to make sure all elevation variations were covered by the placement of GCPs. This was achieved through a basic GNSS based inspection of the terrain; the difference between the highest point on the terrain and the lowest was calculated and divided into seventeen elevation levels. Taking the elevation levels into consideration, the 17 GCP markers were strategically distributed within each level in a 2-1-2 formation as practically as possible. The markers were 40 cm by 40 cm in dimension and had an alternating black/white colour.

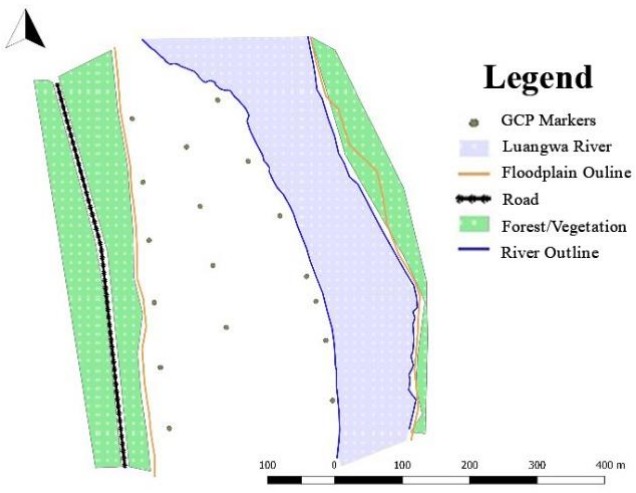

**Figure 4 Spatial distribution of 17 GCPs on Floodplain**

Different GCP numbers and combinations were tested for two different experiments. The first experiment with the objective of determining if open source software could perform as well as commercial software used 3 GCP numbers in a 2-1-2 formation. The 2-1-2 formation is sometimes known as the 'checkerboard' method, it is a relatively common method of distributing marker points on a terrain. The GCP numbers used in this experiment were 5, 9, and 13. The second experiment with the objective to determine the impact of the number and distribution of GCPs used 5 GCP numbers and 2 different formations. The GCP numbers used in this experiment were 0, 5, 9, 13 and 17 GCPs. In the instance where zero GCPs were used, we adjusted the calibration setting to FCP (see introduction section 1) to establish if this would improve results in situations when no GCPs are available. Both the 2-1-2 and the linear biased formations are used in the second experiment. The phrase we refer to as 'linear biased' distribution is a method of marker distribution whereby the markers are placed in a relatively straight line on one side of terrain. In our case the markers are either closer to the river or furthest away from the river (see section 2.3.2).

### 2.2.4 Wet River bathymetry

The Luangwa River, similar to other large tributary rivers of the Zambezi, is perennial meaning the bathymetry of the river needs to be measured under flow conditions. The wet river bathymetry was recorded using a combination of an ADCP and RTK GNSS. The GNSS of the ADCP was not used in favour of the RTK GNSS for improved accuracy. The RTK GNSS was mounted directly onto the ADCP sonar beam, whilst the ADCP was attached to a canoe rowed by local fishermen, as shown in F*igure 5(b)*. The ADCP and the RTK GNSS were configured to take measurements at one second intervals. The canoe

moved from one side to the other in a zigzag manner and tried as much as possible to reach the edges to both sides. The GNSS crossed the river 21 times and a total of 3102 measurements were recorded. The program suitable for the particular ADCP, Winriver II, was used for real-time data collection. For the purposes of interpolation, the canoe was manoeuvred along both
sides of the river. The river was however shallow, especially on the right bank, this means that it was not possible for the canoe to adequately move close to the water line. To capture the slope, the RTK GNSS was mounted on a wooden cart and towed manually along the waterline. An image of the cart is shown in F*igure 5(a)*. The waterline tie line was subsequently used as the true value reference to enable establishment of the level of deviation of the ODM and Agisoft values.

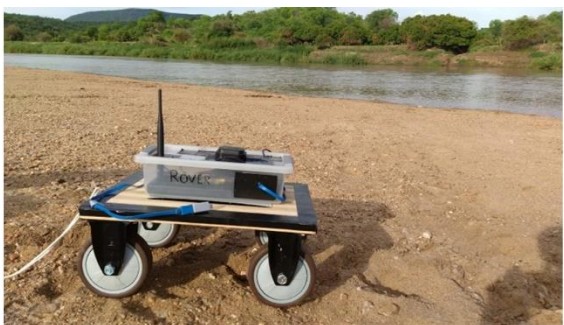 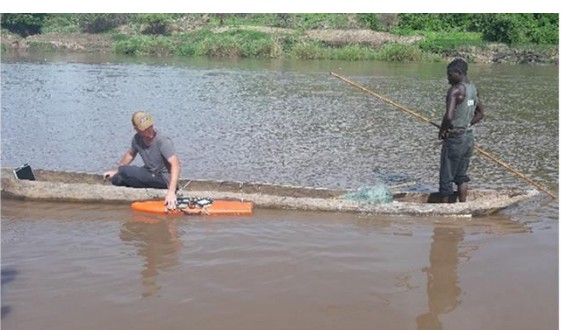

**Figure 5 (a) Low Cost RTK GNSS Rover mounted on a mobile cart for recording RTK water line 5 (b) ADCP combined with an RTK GNSS Rover attached to a fisherman's canoe measuring the wet bathymetry.**

### 2.2.5    Processing the Dry and Wet Bathymetry

Images taken by the UAV were collected and fed into the ODM and Agisoft software. The images were processed locally on
a Dell Core i7 8th generation machine with 32 Gigabytes of RAM. These computer specifications meet the requirements and fit the description of a 'Basic Configuration' (Agisoft, 2021). The same settings were applied in the processing steps as far as was permissible. *Figure 6* outlines the steps which were taken in the production of the point cloud and DEM. The first stage shown on Figure 6 is fieldwork. As with all other images, aerial photographs are optically distorted. In order to correct these distortions, geometric corrections had to be made. These distortions are caused by the camera optics, the topographical relief
and the tilt of the camera (Verhoeven et al., 2013). One of the most effective ways to correct distortions is to make sure that accurate GCPs are recorded and applied. Over and above the traditional GCPs, an RTK water line was measured so as to monitor and correct any potential systematic broad scale distortion (doming effect) which may not have been dealt with by the GCP marker points.

The second stage of the dry bathymetry processing is facilitated by structure from motion (SfM). The constituents that make
up SfM commence with detection of feature points. This is the first step in many computer vision and photogrammetry applications. Despite the existence of approaches which detect edges, ridges and regions of interest, the image features utilised

in most SfM approaches are interest points (IPs). IPs can be defined as the most outstanding locations on an image which are also surrounded by a distinct texture. The following step matches the IPs from one image with the IPs from all other images; the algorithm has to determine which IPs are 2D representatives of the same 3D points. The process of determining the 3D location of interest points using views from different images is called triangulation. The triangulation step requires knowledge of the interior and exterior orientation of images, the output is a sparse point cloud in a local coordinate frame. The final step in SfM which optimises the sparse 3D structure and the projection matrices simultaneously through a robust iteration is called bundle adjustment.

The third stage commences with the application of a coordinate reference system (geo-referencing) to the model. This step is necessitated by the inherent scale ambiguity of the SfM output. This is to say that if the sparse 3D structure is scaled by an arbitrary value and the distance between the camera's positions are simultaneously scaled by the same factor, then the structure will remain the same. The two main methods are either to import at least 3 well distributed GCPs and transform the complete model, or to import at least 3 accurately known camera positions or GCPs and use them as constraints during bundle adjustment (Barazzetti et al., 2012). The next step is Multi View Stereo (MVS), which facilitates the creation of a dense point cloud. This MVS algorithm uses information on the orientation of images to compute a dense structure. This is possible because the outputs are pixel based as opposed to feature point based.

The final stage facilitates the creation of a Digital Elevation Model (DEM) and the orthomosaic. The orthomosaic is important for visualisation of the terrain at a high resolution. This makes it possible to calculate the RMSE of GCPs which would otherwise be difficult to identify.

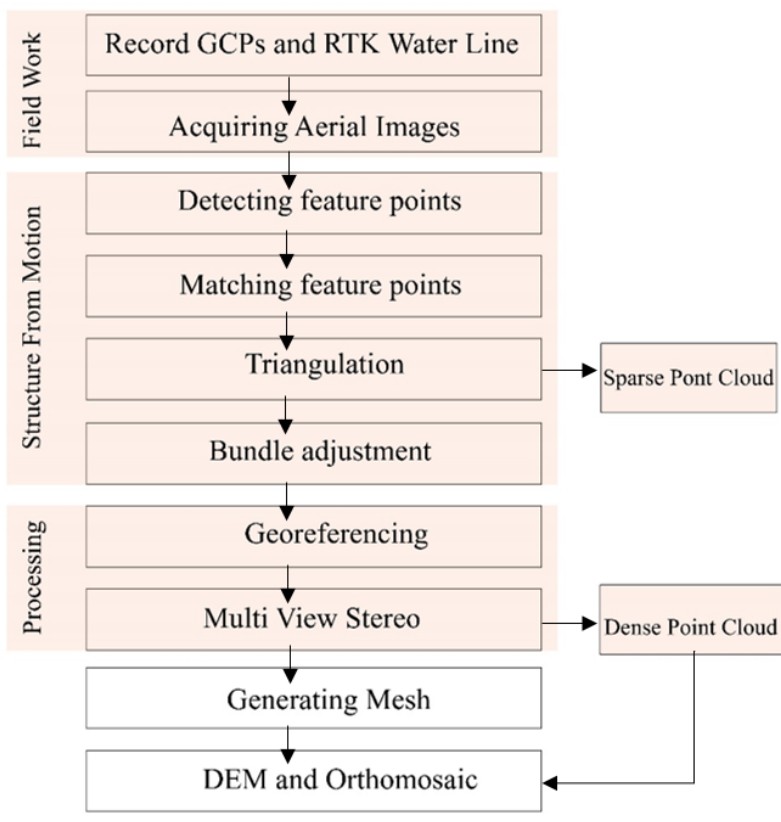


**Figure 6 Photogrammetry Process from image collection to reconstruction of elevation model adapted from Balogh and Kiss (2014)**

The processes summarised in Fig. 6 (third block until the last block) was repeated four times with different sets of GCPs each time (5, 9, 13, 17 GCPs) for both software packages. The Agisoft software version 1.5.1 reconstruction took approximately 9

hours to process each set of images whereas, ODM took 2 hours.

The wet river bathymetry point cloud is processed as follows. Each measurement point taken on the river consists of the attributes depth (measured with the ADCP), latitude, longitude and height (measured with the RTK GNSS). The depth measurement is subtracted from the water height to acquire the bed level, and combined with the longitudinal and latitudinal coordinates. Before the wet bathymetry is merged to the dry bathymetry, the wet river transects have to be volumised. This

process entails conversion of the sparse point cloud made of transect points into pixels through linear interpolation with the nearest non-empty cell.

In order to obtain the full bathymetry of the river, the dry bathymetry and the wet bathymetry are merged together in the software. In occurrences whereby there are overlaps or edges we choose to treat these through linear interpolation as well.

After merging, three cross sections perpendicular to the river were extracted such that a relationship between area and perimeter
could be established over the entire cross-section, including both wet and dry bathymetry.

### 2.3 Reconstruction Experiments

#### 2.3.1 Impact of the used processing software

A relatively simple experiment to judge if ODM can be used as a viable alternative to costly proprietary software was employed. The experiment sought to validate the accuracy of open-source software versus commercially available software
by comparing ODM (open-source) with Agisoft Metashape (commercial), respectively. The availability of GCPs made this possible. We considered the root mean square error (RMSE) of checkpoints. RMSE metric is widely employed as a measure of conformity between two DEMs (Alidoost and Arefi, 2017). If the RMSE values are of comparable nature, comparing one package against the other (magnitude, distribution, presence of outliers) then they perform similarly. To calculate these RMSE values, only those reference points which were not used in the reconstruction were made use of, this allowed for an independent
estimation made by both software packages. The RMSE was computed using *Equation 2 and Equation 3* in the horizontal and vertical direction respectively.

$$RMSE_{xy} = \sqrt{\frac{1}{n}\sum_{i=1}^{n}(\Delta X_i^2 + \Delta Y_i^2)} \qquad (2)$$

$$RMSE_z = \sqrt{\frac{1}{n}\sum_{i=1}^{n}(\Delta Z_i^2)} \qquad (3)$$

Where          $\Delta X_i$ = residual of the [i]th value in the x axis

$\Delta Y_i$ = residual of the [i]th value in the y axis

$\Delta Z_i$ = residual of the [i]th value in the z axis

340                    n = number of check points (GCPs that were not used in the reconstruction)

DEMs based on five, nine, thirteen and seventeen GCPs were exported from ODM and Agisoft. The DEMs were fed into the Geographic Information System (GIS) QGIS and a point sampling tool was used to extract elevation values at the corresponding coordinates of the GCPs that were not used in the reconstruction. This ensured that an independent estimate of the RMSE could be established. A bootstrapping experiment was conducted on the errors of the individual GCPs that were
used to calculate the RMSE. This experiment was performed to test the stability of the RMSE. In the experiment random samples of error were drawn from the 5, 9 and 13 GCPs. The sampled errors, which were equal in number to the available GCPs, were sampled with replacement to obtain new RMSE values. The process was then repeated for 1000 drawn sample sets. Given that this first experiment led to the conclusion that ODM is a satisfactory choice and it is free and Open-Source (see Section 3.1) the remaining experiments were only conducted with ODM.

       **2.3.2 Impact of GCP placement and density on accuracy**

This experimental objective was divided into two parts. The first was to establish the impact of GCP density on DEM accuracy. The second part was to establish the impact of placing GCPs further or closer to the flowing river. In both instances a comparison of absolute error was made with the RTK tie line which was acquired using the RTK GNSS mounted on a mobile cart. The Python package 'rasterio' was used to extract elevation values at corresponding coordinates. For the first part, elevations from the DEMs with 5, 9 13 and 17 GCPS were extracted and compared to the RTK line elevations. For each reconstruction, the maximum number of checkpoints available were used to verify the results. The sum total of GCPs used in each reconstruction and checkpoints was always equivalent to the total number of GCPs available (17). The reconstruct with 5 GCPS, had 12 checkpoints, the 9 GCP reconstruction had 8 checkpoints, the 13 GCP reconstruction had 4 checkpoints and the 17 GCP reconstruction essentially had 0 checkpoints available. The exact same GCP numbers and distributions were used for the reconstruction in both Agisoft ODM. *Figure 7* shows the locations and particular markers which were selected for each set of GCPs. The GCPs were placed in a 2-1-2 formation which took into account the range of elevations as much as possible.

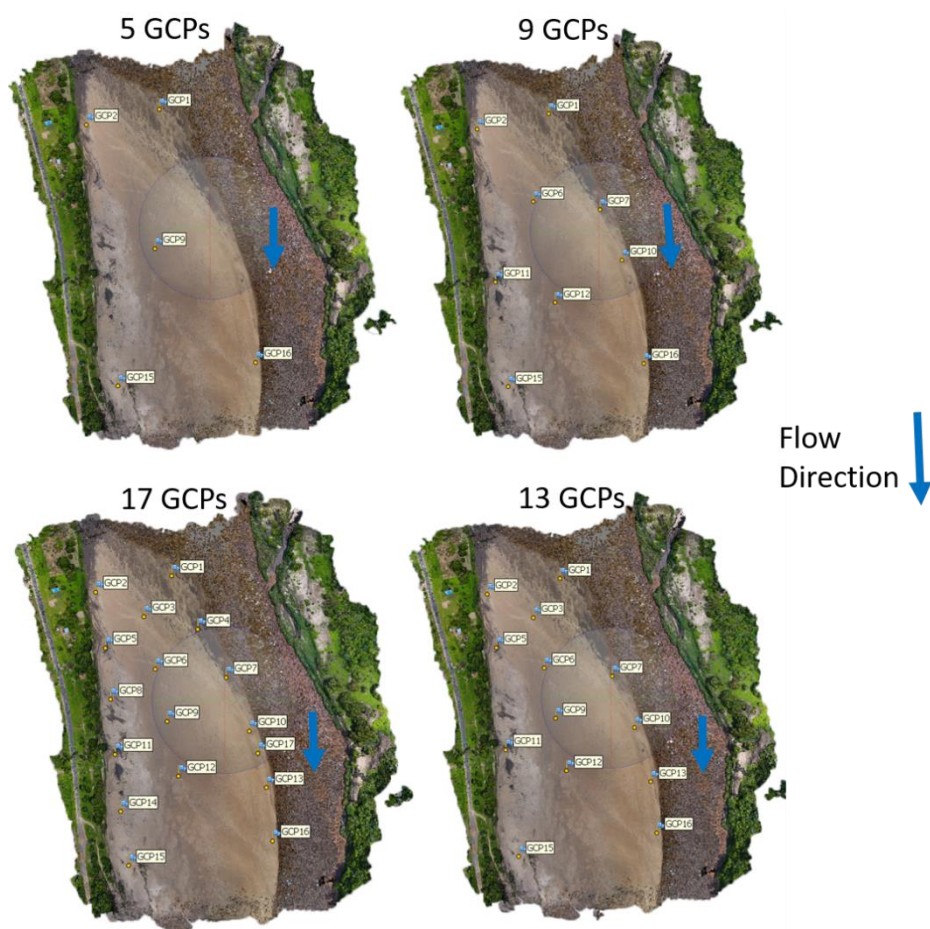

**Figure 7 GCP marker distribution along the floodplain**

For the second part, many studies have indicated that photogrammetry is incapable of adequately mapping a flowing river
because it reflects light (Bandini et al., 2017; Dai et al., 2018). The noise generated on the river surface has a negative impact
on the overall accuracy of the DEM. In order to establish the significance of this noise, elevation extrapolations from the DEMs
constructed using 9 GCPs closest to the river and 9 GCPs furthest from the river were compared. The GCP markers are placed
in linear biased manner parallel to the RTK reference line. *Figure 8* shows the positions of the GCPs placed further and closer
to the river. The figure also shows an orthophoto to be able to identify the river's water surface and other features such as the
vegetation on the natural levee of the river's floodplain.

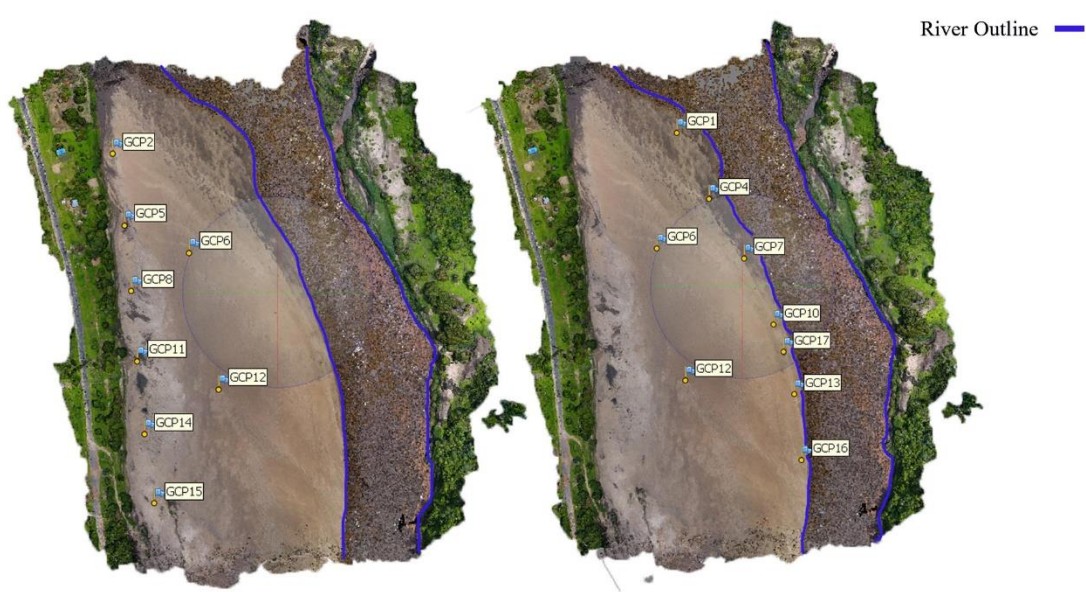

**Figure 8. GCP marker distribution closer and further from the river and wetted river perimeter measured during the survey, projected on an orthophoto result**

### 2.3.3 Impact of DEM variations on hydraulic conveyance

375 We investigated how variations in DEM reconstruction choices impact on conveyance characteristics. We determined conveyance versus depth relationships over several cross-sections in each DEM created. This was done for all the elevation models generated using a different number of GCPs so that the established relationships could be compared. *Figure 9* shows the location of the cross sections which were extracted from the respective reconstructions.

In addition, we compared DEM derived hydraulic slope with an independent estimate of slope using the in-situ RTK GNSS
380 tie line (see Section 2.2.3 for a description of the acquisition method). The first method calculated slope entirely based on an independent reference tie line. In order to attain the actual elevation values, the height of the cart and the container were subtracted from the height measurements. The plot consists of 898 measurement points with a standard deviation of 0.018 metres. A regression line was fit through the data and the waterline slope was determined to be 0.000230. A plot of the regression line is shown in Appendix B3. The second method involves the extraction of the slope from the terrain outputs
385 produced by the photogrammetry process. The sample method of the Rasterio library in Python has been used. This method sampled the closest point to every coordinate in the RTK track. Thereafter, a regression line was fit through the various elevations so as to determine the slope of the various photogrammetry outputs. The outputs were then compared, taking the slope derived from the GNSS as the true value.

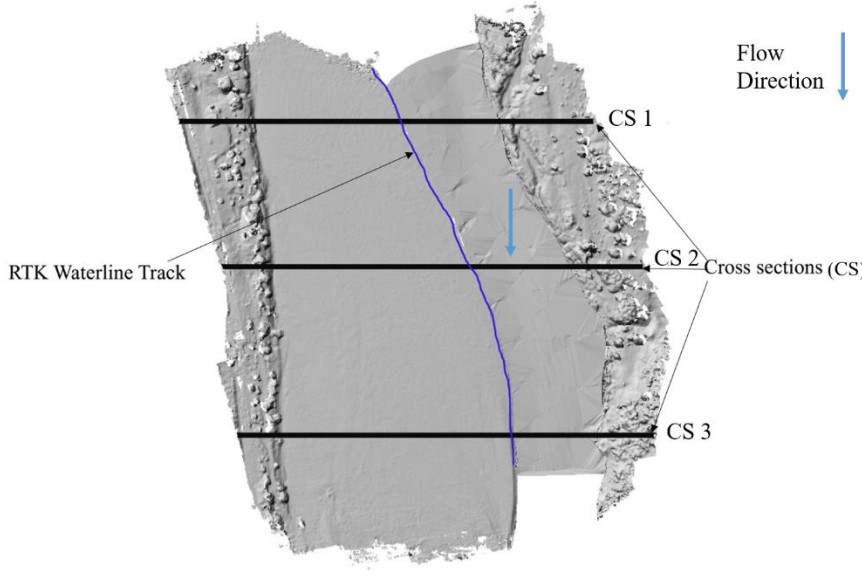

**Figure 9 The location of 3 cross sections (CS1, CS2 and CS3)**

## 3    Results

In summary, the assessment of the impact of processing methods on quality of terrain data, focussing on geometry of hydraulic properties consisted of three steps: applicability of open source versus proprietary photogrammetry software, the impact of GCP density and placement on DEM quality, and the impact of variations in DEMs on conveyance and slope. In this section, we present the results of these three steps.

### 3.1    Impact of the used processing software

In order to assess the applicability of open-source software the RMSE of terrain models processed in ODM were compared with those from Agisoft Metashape. The results are presented in *Table 1*.

**Table 1 RMSE of different GCP combinations**

|  | Agisoft | | ODM (m) | |
|---|---|---|---|---|
| Configuration | Horizontal RMSE [m] | Vertical RMSE [m] | Horizontal RMSE [m] | Vertical RMSE [m] |

| | | | | |
|---|---|---|---|---|
| **5 GCPs** | 0.415 | 0.594 | 0.686 | 0.592 |
| **9 GCPs** | 0.259 | 0.290 | 0.406 | 0.344 |
| **13 GCPs** | 0.300 | 0.395 | 0.431 | 0.380 |

The results indicate Agisoft RMSE values that are comparable to those calculated when ODM was used for reconstruction. The two software products generally follow a trend whereby increasing the number of GCPs from 5 to 9 results in a notable decrease in RMSE. A further increase from 9 to 13 GCPs results in an increase in RMSE. This result is counter intuitive, however, given that the error was calculated based on GCPs which were not used in the reconstruction, it follows that increasing the number of GCPs simultaneously decreased the sample size available for error calculation. A reduced sample size meant that outlier error values may well result in a poorer resultant RMSE. In general, the RMSE values of Agisoft and ODM were similar, however, we note that the sample size of data used to calculate the RMSE was not large enough to provide statistical confidence. To that end, a bootstrapping experiment was conducted to establish if there was a significant similarity in the performance of ODM in comparison to Agisoft (see Section 2.3.1). The bootstrapping experiment is particularly appropriate for small sample sizes and data sets which do not necessarily follow a normal distribution (Freedman, 2007). The results of the bootstrap experiment are presented in *Figure 10*.

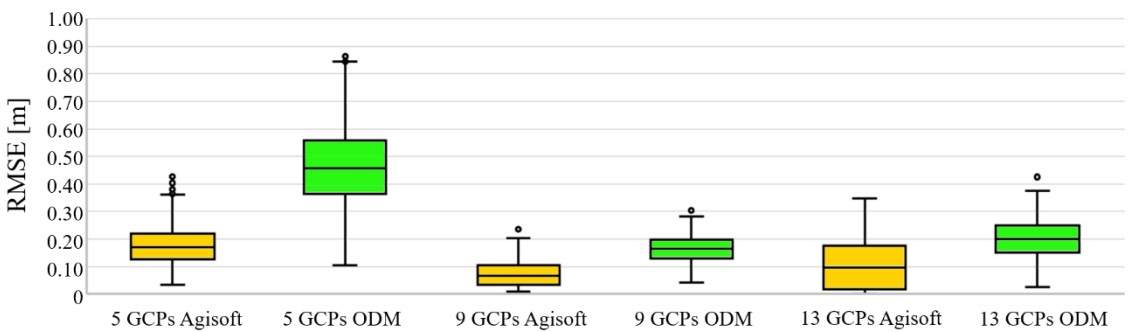

**Figure 10. Bootstrap Box plot experiment comparing the performance of ODM against Agisoft for varying number of GCPs.**

Using 5 GCPs, there is a relatively large difference between the RMSE of Agisoft and ODM. We attribute this difference to the inherent capacity of Agisoft to perform better than ODM in instances where there are few control points. The graph suggests that, out of the selected number of comparisons, 13 GCPs is the optimal balance between GCPs that correct the reconstruction and checkpoints to calculate the RMSE. The representation indicates a strong resemblance between errors in ODM and Agisoft. The overlapping box plot in the 13 GCP configuration affirms the comparability of the products. However, a notable downside

of ODM is indicated by the RMSE which is twice that of Agisoft. Despite this downside, the absolute RMSE error is limited
to less than 0.20 m, which is acceptable for the purposes of merging with wet bathymetry. The results confirm the potential application of open-source software as an alternative for commercial options without significant compromise on accuracy. Accordingly, the remainder of the results are processed and analysed based on the ODM software package.

## 3.2    Impact of GCP placement and density on accuracy of hydraulic features

The aim of this experiment was to assess the impact of variations in the number of Ground Control Points (GCPs) and the
distribution of the GCP markers on the quality of DEMs, with a particular emphasis on characteristics that impact on hydraulics. Five different GCP numbers (0, 5, 9, 13, and 17) and two specialised settings (Brown-Conrady and Fixed camera parameter) were compared. We observed dome-like deformations in all of the elevation extractions. This phenomenon, known as the 'doming effect' (also known as "bowling effect", described in section 1) is exemplified in *Figure 11*. The effect is apparent despite attempts to avoid the aforementioned phenomena through deliberate flight practices such as a 10° camera
angle and a 20° alternating flight path.

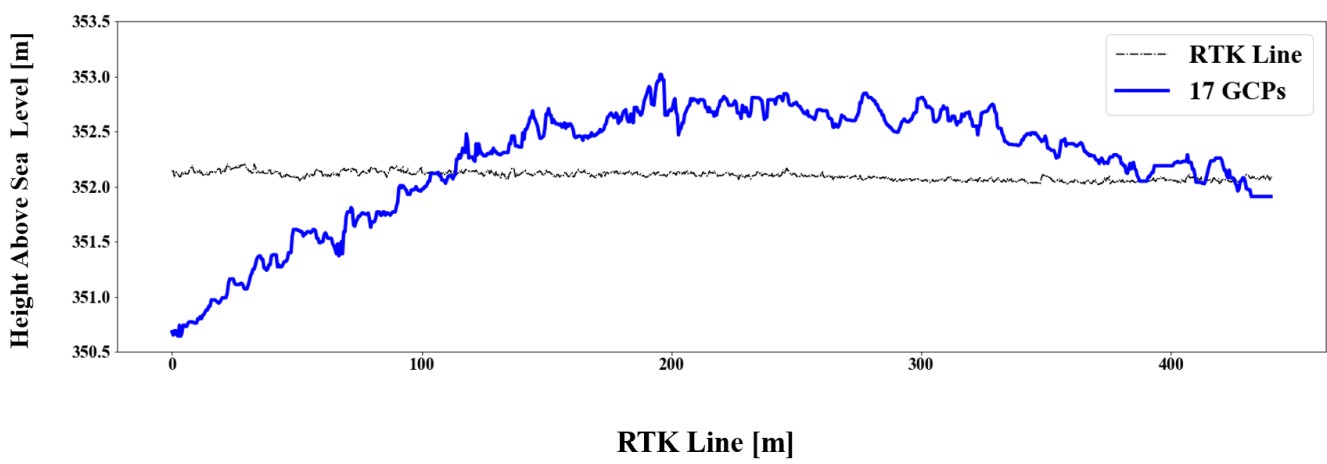

**Figure 11. The 'Doming effect' visualised through comparison of elevation levels extracted from the RTK line vs elevation values extracted from the reconstructed photogrammetry-based point cloud.**

A rather practical approach was used to correct for the doming effect. A first order polynomial was fitted through the RTK GNSS track. A second order polynomial was then fitted through all the reconstructed point clouds. The error was then determined by calculating the absolute difference between the two polynomials for the given length. The respective clouds were divided into 1500 sections from north to south whereby every point within each section was assumed to be deformed by

the same elevation value. The absolute errors were then applied as corrections to the point clouds depending on which section
each location fell in. F*igure 12* shows corrections made to the reconstruction based on 5 GCPs. Appendix B shows the
corrections which were performed on all other terrain models.

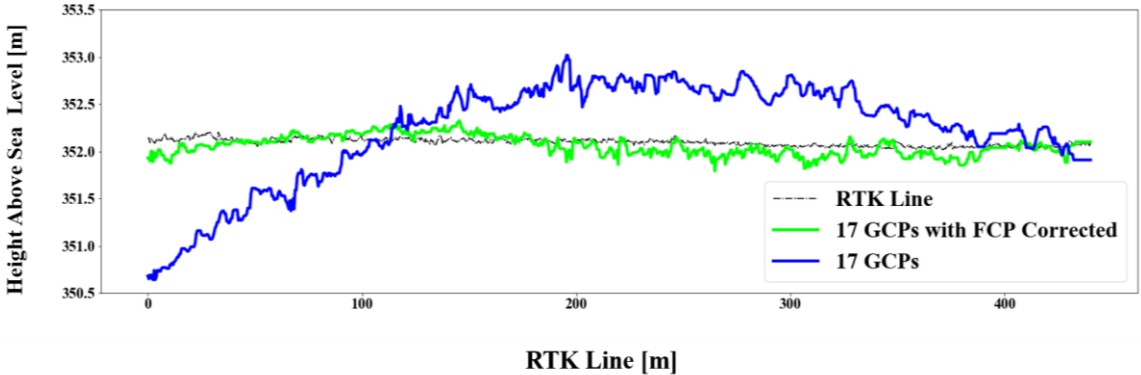

**Figure 12. A visualisation of the effect of correcting the doming effect**

The assessment was conducted based on the RTK waterline track and the results are presented in *table 2*. The results indicate
a decrease in the RMSE as we increase the number of GCPs. However the incremental benefit of increasing the number of
GCPs beyond 5 becomes smaller as more control points were added to the reconstruction. Noticeably, the RMSEs derived
based on the GCP checkpoints was similar to that which was obtained based on the RTK waterline as a reference. This implies
that the RTK waterline track is a potential substitute when calculating the error in a photogrammetry reconstructed model. The
RMSE values derived from the 'No GCPs' and from using the 'Brown-Conrady' configuration showed significant inaccuracy
and therefore rendered inapplicable. However, the 'Fixed Camera Parameter' configuration performed reasonably well (RMSE
= 0.618m), considering no control points were used.

**Table 2 RMSE of different GCP combination and configurations**

| Configuration | $RMSE_z$ [m] |
|---|---|
| | Based on RTK track |
| 5 GCPs | 0.558 |
| 9 GCPs | 0.581 |
| 13 GCPs | 0.486 |

| | |
|---|---|
| 17 GCPs | 0.479 |
| FCP | 0.618 |

We identified a bias in terms of the errors calculated when GCPs are closer to or further from the river. The results are presented
in *table 3*. Similar to the aforementioned experiment, the RTK track was used as a reference. The RMSE is less when GCPs
closer to the river (approximately 20 m away) are used in the reconstruction than when GCPs further away are used. We
hypothesize that the GCP distribution used in the experiment 'Closer to River', is such that GCPs are placed much closer to
the reference line, therefore better conditioning the part of the reconstruction close to the RTK track. Our hypothesis is
reaffirmed by the results of calculating the RMSE based on GCPs as shown in table 3. Similarly, the RMSE is less when GCPs
used in the reconstruction are closer to the River.

**Table 3 RMSE comparison further and closer to the river**

| Configuration | ODM RMSE error | |
| | RMSE [m]<br>Based on RTK line | RMSE [m]<br>Based on GCPs |
|---|---|---|
| Closer to River | 0.374 | 0.242 |
| Further from River | 0.771 | 0.926 |

### 3.3 Impact of DEM variations on hydraulic conveyance and slope

Hydraulic conveyance was computed from the merged dry and wet bathymetry. We performed a comparison of the hydraulic
conveyance across various reconstructions. Furthermore, we compared the hydraulic slope of the various reconstructions with
an independent slope estimate measured from an in-situ RTK GNSS tie line. In order to extract the cross-section elevations,
the full bathymetry of the river had to be utilised. *Figure 13* is a visualisation of the process of generating a volumised wet
bathymetry from separate components. The wet river point cloud, shown in *Figure 13*, covers 555 metres of the river length
and consists of 5,164 points. The latitude and longitude originate from RTK GPS measurements whereas the height component
is determined using both RTK GNSS and an ADCP as described in section 2.2. The maximum and minimum height of the
point cloud are 352.20 and 348.45 metres respectively.

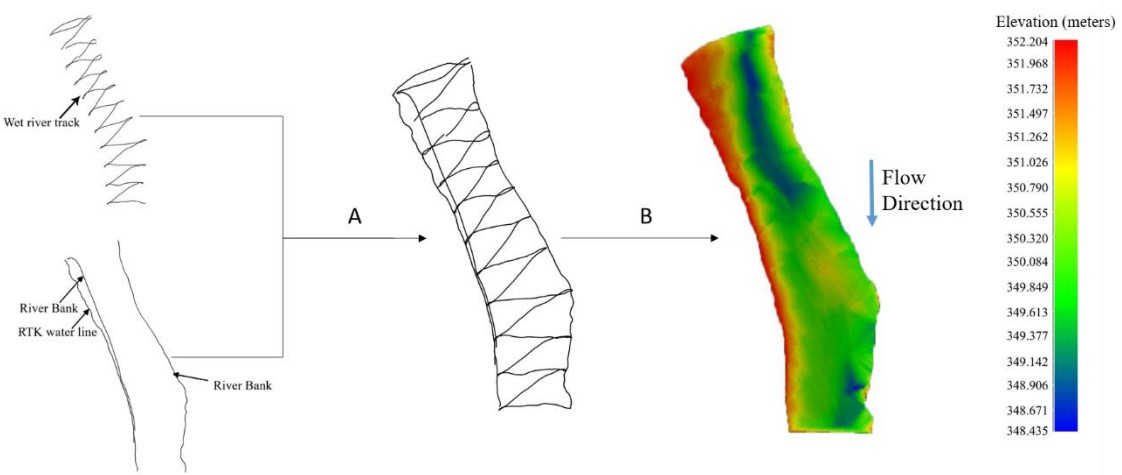

**Figure 13. Wet Bathymetry processing (A: Merging the wet bathymetry transects which were measured using the ADCP with the RTK line as well as the boundary B: Volumisation of the merged products achieved through linear interpolation).**

The dry river bathymetry is constructed using photogrammetry and RTK GNSS as described in section 2.2. The various point clouds represent an area of approximately 679 x 551 metres. Like the wet river, each point contains a latitude, longitude and height component with a maximum and minimum height of 383.4 (hill in the south east corner) and 350.2 metres respectively.

In order to extract the cross-sections, the dry and wet bathymetry had to be merged and subsequently volumised. These two processes that were conducted in Cloud Compare are exemplified in *Figure 14*.


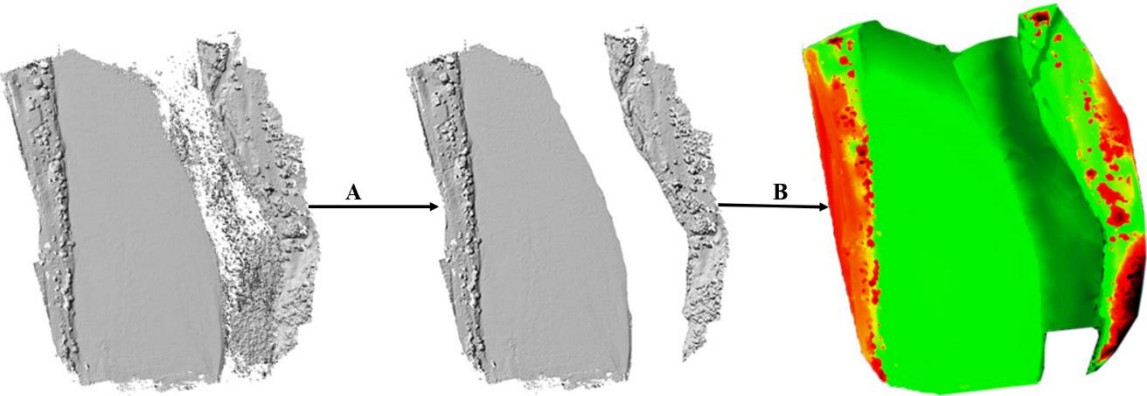

**Figure 14. Floodplain processing (A: extraction of water surface river section, B: Merging dry bathymetry with wet bathymetry and Volumisation).**

*Figure 15* shows an extraction of the cross section on the northern side of the terrain model (CS1). The GCP configuration
with 5, 9, 13 and 17 GCPs present very similar cross-sectional properties. The results are similar for all cross sections
(Appendix B4). The configuration with no GCPs and Brown -Conrady significantly underestimated the actual height by
approximately 13 meters.

In an attempt to improve the results when no GCPs are available, we applied a configuration setting known as FCP (Fixed
Camera Parameters). The FCP results showed a significant improvement, the shape of the cross section was similar to the
experiments with GCPs though visibly below the rest.

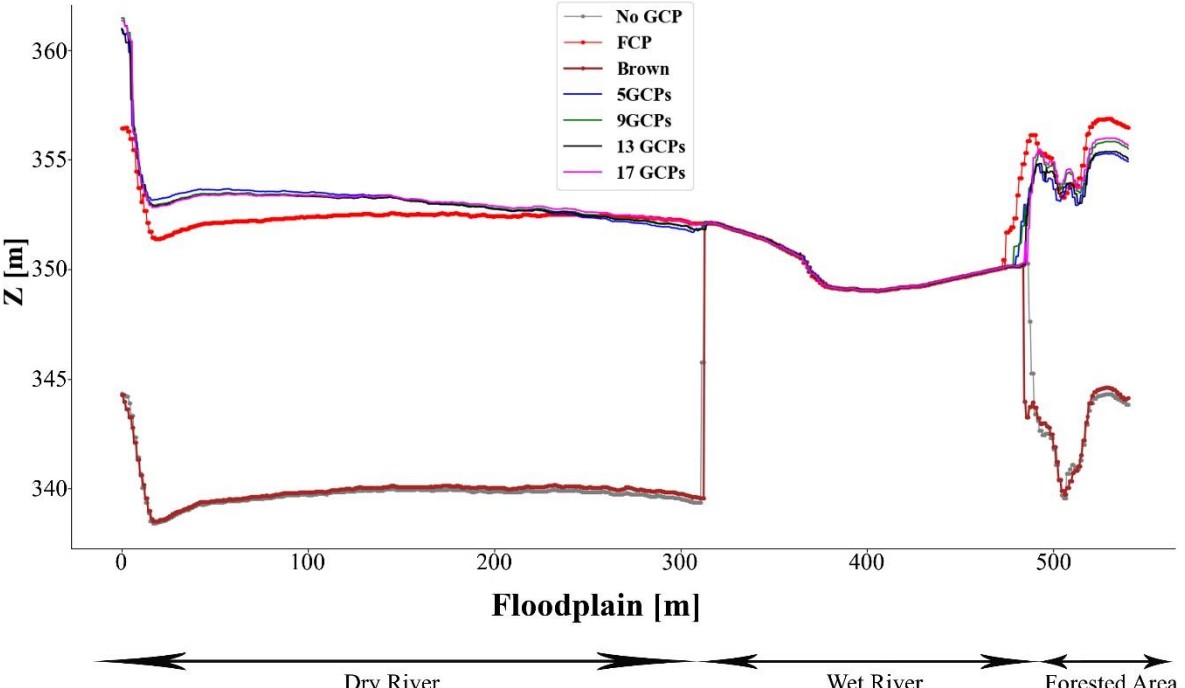

**Figure 15. Extract of Cross section 1 combined geometry of Dry and Wet bathymetry**

The hydraulic conveyance estimation graph is presented in *Figure 16*. As anticipated, results indicate no significant difference
among conveyances estimated based on 5, 9, 13, and 17 GCPs. The conveyances estimated based on the 'no GCP' and 'Brown-
Conrady' configuration are not meaningful because of the clear offset between the photogrammetry results and the RTK
results. The conveyance based on the FCP performed better than Brown Conrady and no GCP configuration. However, the
estimated conveyance was significantly different from the conveyances estimated using GCPs. The results were similar for all
3 cross sections (Appendix B). The left bank of the river seems significantly higher than the right bank. This is due to the
riparian vegetation, present on the left bank. This made access using the canoe (thus ADCP) difficult. Furthermore, the canopy
cover from the riparian vegetation made it impossible for the photogrammetry to resolve the ground surface here.

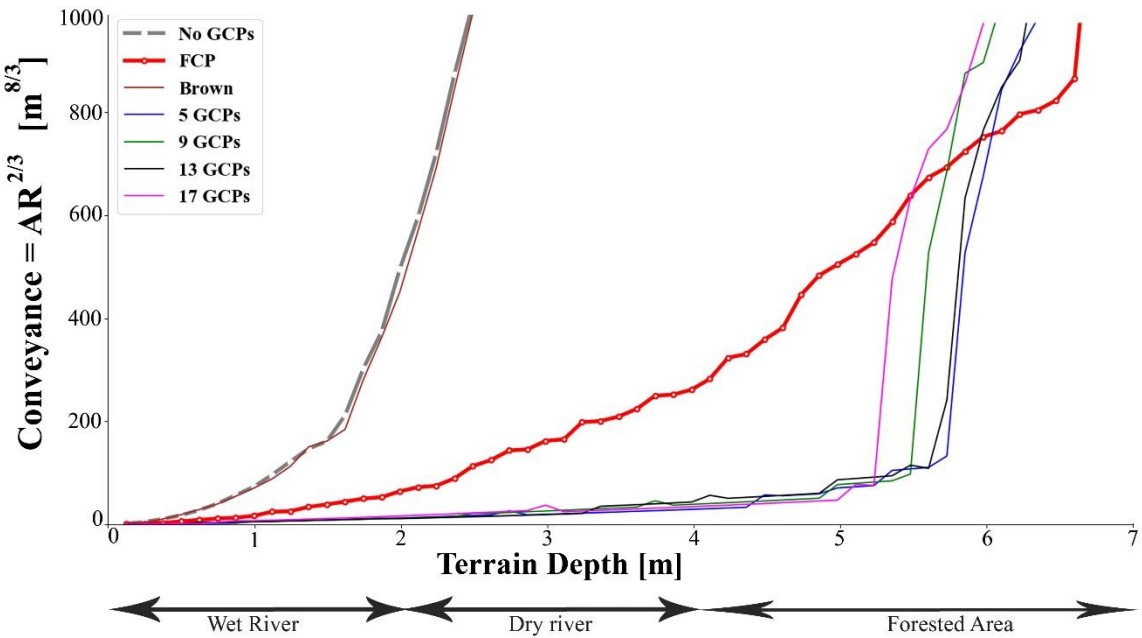

**Figure 16 Cross section 3 (south of the terrain) conveyance vs depth relationship**

The slope calculations, shown in *Table 4,* reveal a significant difference between the true slope (RTK GNSS), and the photogrammetry derived slope values. This is despite a correction of the doming effect as described in *section 3.2.* Among photogrammetry based slope derivations, there were relatively large variations. Results indicate that for the purposes of hydraulic rating, slope derived from SFM is inapplicable due to high levels of inaccuracy.


**Table 4 Slope estimations**

| Configuration | Hydraulic slope [ $* 10^{-4}$ m] |
|---|---|
| RTK GNSS Track | -2.300 |
| 5 GCPs | -3.935 |
| 9 GCPs | -3.286 |

| | |
|---|---|
| 13 GCPs | -3.749 |
| 17 GCPs | -3.891 |
| No GCPs FCP | -3.995 |

## 4    Conclusions and Recommendations

This study reinforced the capability of low-cost instruments, such as UAVs in combination with RTK GNSS, being applied to
perform physically based remote river rating. The performance of the open-source photogrammetry software substantiated the
claim that, free and open-source available packages, are capable of producing results which are as good as proprietary
alternatives as shown by the RMSE analyses. Across different GCP distributions, no significant difference was observed
between the errors calculated based on open-source software and those calculated based on commercial software packages.
This, combined with the fact that a UAV data can be acquired relatively quickly and would be affordable to many water
management institutions in low income economies opens doors for use in low resource settings. Apart from cost implications,
the open-source software provided an option in the form of a 'fixed camera parameter' configuration which significantly
reduced the RMSE of the reconstruction, even without the use of GCPs. The results had limitations in terms of the sample size
used for calculating the RMSE of the GCPs. For instance, when reconstruction was performed based on 13 GCPs, only 4 GCPs
were available to use as validation points. In future studies, it would also be useful not only increase the number of independent
checkpoints but to also measure the RTK track further away from the river to avoid influence of poor river photogrammetry
reconstruction.

As anticipated, increasing the number of GCPs had an inverse effect on the RMSE. However, the gradual improvement in
accuracy of the reconstruction diminished disproportionately. For the selected trials, a reconstruction based on 13 GCPs
provides the most accurate RMSE results. It provides an optimal balance between the number of GCPs for reconstruction and
the number of validation points. In addition, we note that accuracy cannot be determined based on GCP density alone. The
distribution of GCPs proves to be as critical as the GCP density in order to achieve optimal accuracy. In certain cases, priority
must be placed on the GCP distribution so that the output is representative of a wider range of elevation values.  Placing more
GCPs in proximity to potentially problematic areas such as forests or water significantly improves the overall output of the
reconstruction.

The effective impact of variations in GCPs on geometry is realised in the form of conveyance. Despite the optimal number of
GCPs being thirteen (13), the study concludes that five (5) GCPs evenly spread out across a floodplain of approximately 40
hectares and flying at an elevation of 100 m is sufficient to generate an elevation model that meets the requirements of accurate

conveyance estimation. Configurations such as the FCP advance the model reconstruction but do not achieve satisfactory accuracy without GCPs. Slope estimation based on photogrammetry reconstructions was not satisfactory under any GCP configuration tested. The novel method of measuring an RTK GNSS line is therefore a critical alternative to establish the slope by correcting for the doming effect. Furthermore, we note that conveyance is more impacted by the quality of the wet bathymetry collected by the GNSS than by the dry SfM bathymetry. Therefore, careful attention needs to be paid into making sure the wet bathymetry is measured as accurately as possible.

A novel approach to generate a seamless bathymetry through merging and volumisation was successfully tested. However, there was visible evidence of some mismatch in elevation, particularly on the upper part of the study area where the wet and dry bathymetry were merged. To counteract this discrepancy, future studies may consider increasing wet bathymetry transects such that algorithms used to merge the two bathymetries (in this case Cloud Compare) have access to more transects. The additional transects would improve the quality of the wet bathymetry constructed through linear interpolation. Furthermore, results presented here encourage future studies to investigate the impact of variations in the number of GCPs on discharge estimations in a hydraulic model with different hydrodynamic boundary conditions. Within the envisioned hydraulic model it would be important to extend terrain downward to reduce backwater effects.


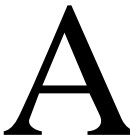

**4.1    Data Collection**

This appendix contains figures, tables and photo which complement the data collection method.

Figure A.1 shows the components and setup of the constructed Real Time Kinematic GNSS. The container on the right-hand

side in Figure A.1 contains the base, the other container contains the rover. Both containers include two SimpleRTK2B boards

with a u-blox-ZED9P module, a Raspberry Pi, two GNSS antennas, an XBEE shield and a long range radio antenna. With this

hardware, two complete RTK GNSS sets can be constructed, one based on long range radio communication and one based on

a 4G internet connection. The SimpleRTK2B board with the XBEE shield works with the radio module and is used during the

fieldwork.

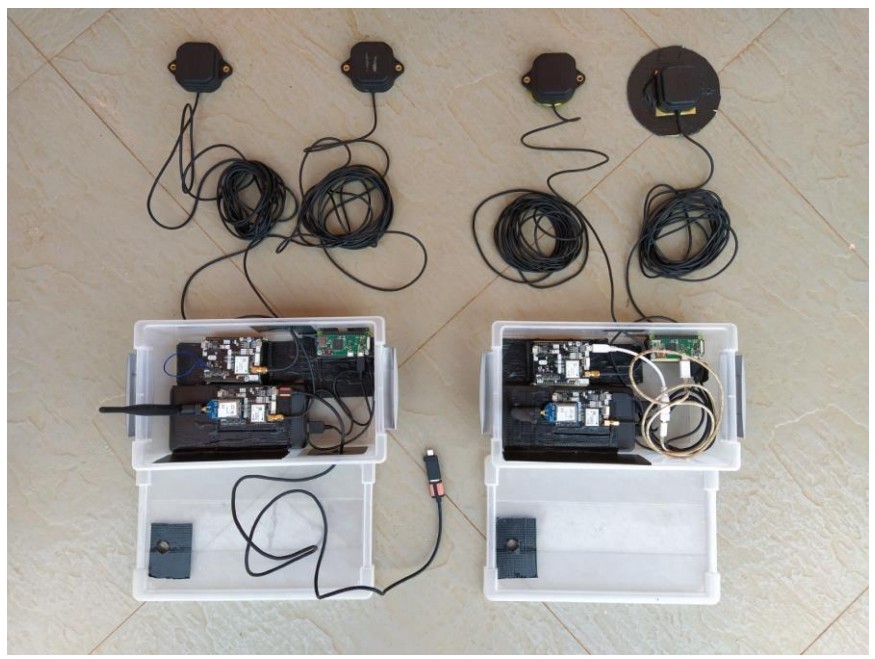

**Fig. A1. 2 sets of RTK GNSS Equipment, one base of a long range radio connection and one based on 4G connection.**

Figure B.1 shows the bathymetric data collection setup with the ADCP tied to the wooden canoe of a local boatman. On top

of the sonar an RTK GNSS receiver is mounted which is, via a SimpleRTK2B board, connected to a smartphone logging the

location measurements with a one second time interval. The ADCP is connected to a laptop running Winriver II which stores the depth measurements. Figure B.1: The ADCP connected to the canoe with the GNSS receiver mounted on top of the sonar.

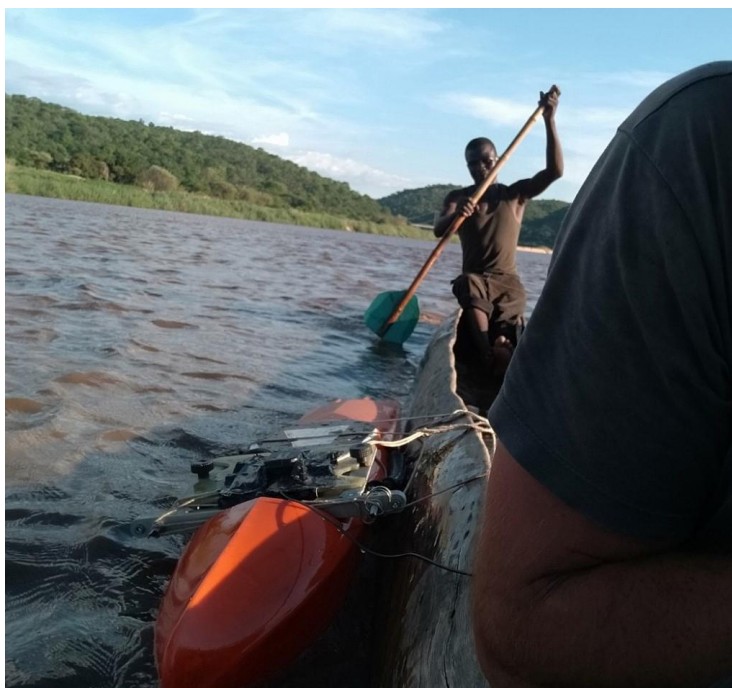

**Fig. A2. ADCP with a Low cost RTK GNSS attached to canoe**





# B


## 4.2 Wet and Dry Bathymetry

*Figure B1* shows the 'bowling' or 'doming' effect on terrain models. The top left graph represents the relationship between height and track for the 5 GCP terrain. The centre left graph represents the relationship between and track for the 5 GCP terrain after FCP correction. The bottom left graph represents the relationship between and track for the 5 GCP terrain after both FCP

and doming correction. The top right graph represents the relationship between height and track for the 9 GCP terrain. The centre right graph represents the relationship between and track for the 9 GCP terrain after FCP correction. The bottom right graph represents the relationship between and track for the 9 GCP terrain after both FCP and doming correction.

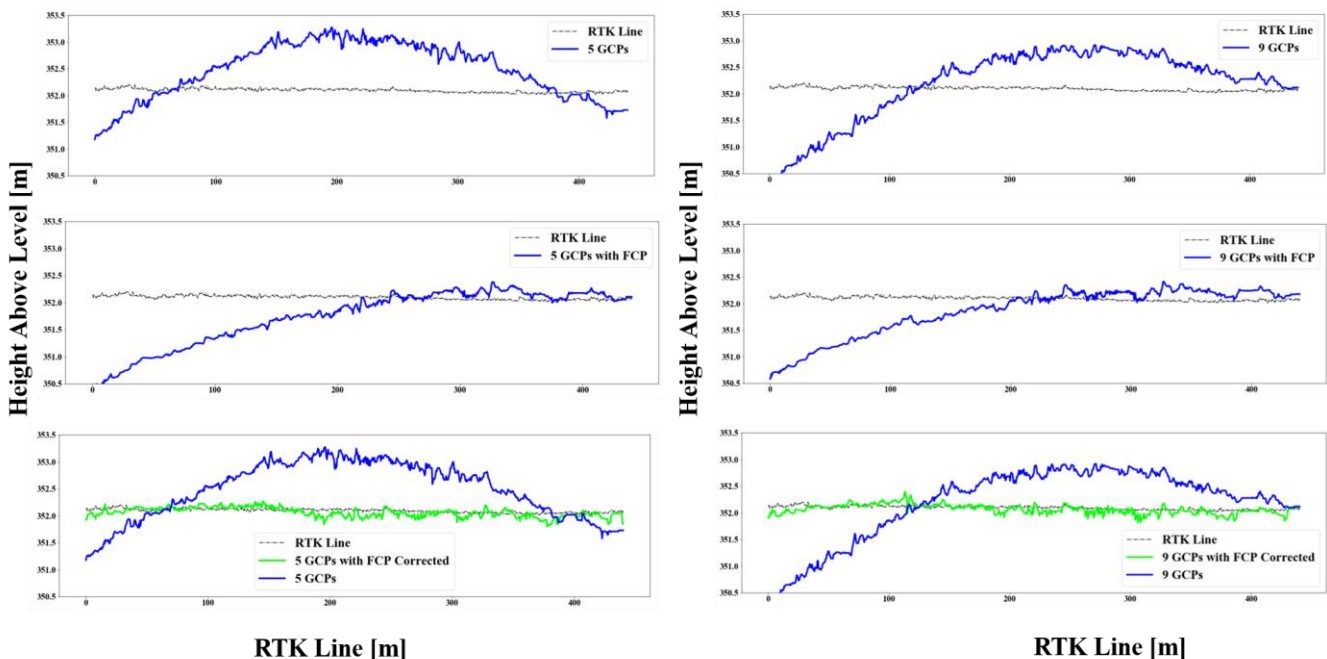

**Fig.B1. Correction for the doming effect**

Figure B2 shows the 'bowling' or 'doming' doming effect on terrain models. The top left graph represents the relationship between height and track for the 13 GCP terrain. The centre left graph represents the relationship between and track for the 13 GCP terrain after FCP correction. The bottom left graph represents the relationship between and track for the 13 GCP terrain after both FCP and doming correction. The top right graph represents the relationship between height and track for the 17 GCP

 terrain. The centre right graph represents the relationship between and track for the 17 GCP terrain after FCP correction. The bottom right graph represents the relationship between and track for the 17 GCP terrain after both FCP and doming correction.

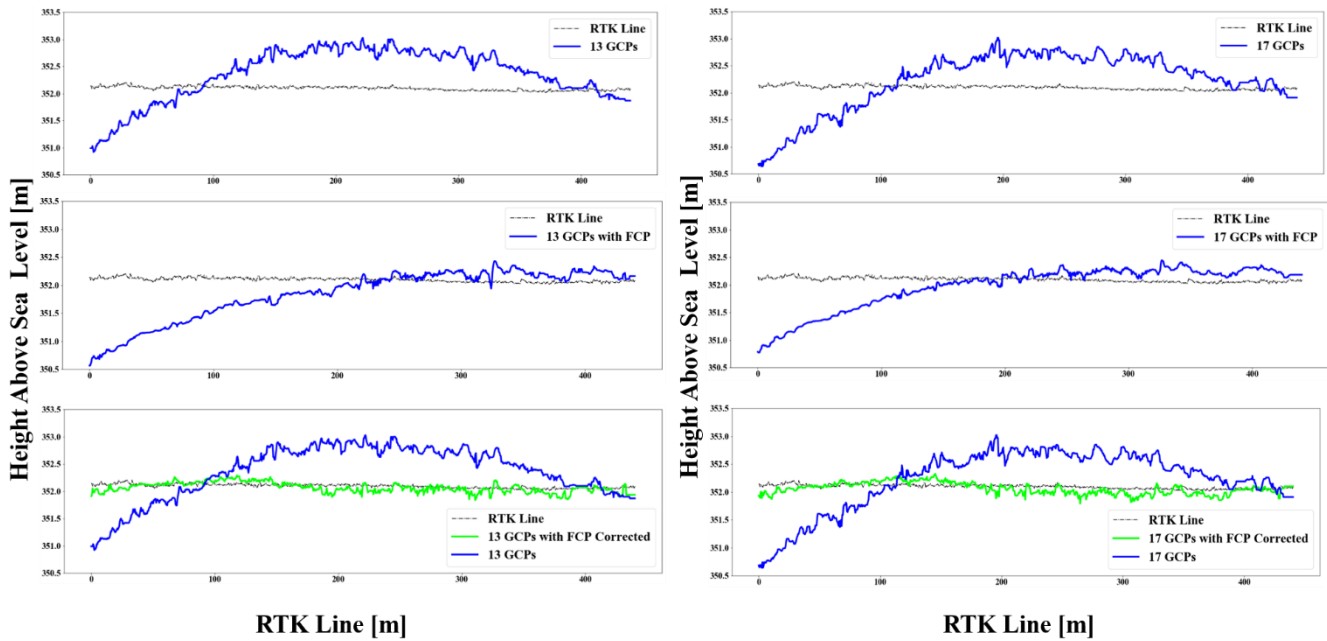

**Fig. B2. Correcting the doming effect**

Figure B3 shows the regression line fit through extracted tracks lines. The top left graph represents the relationship between

 height and track for the RTK track. The centre left graph represents the relationship between height and track for the 9 GCP. The bottom left graph represents the relationship between height and track for the 17 GCP terrain. The top right graph represents the relationship between height and track for the 5 GCP terrain. The centre right graph represents the relationship between height and track for the 13 GCP. The bottom right graph represents the relationship between height and track for the no GCP terrain

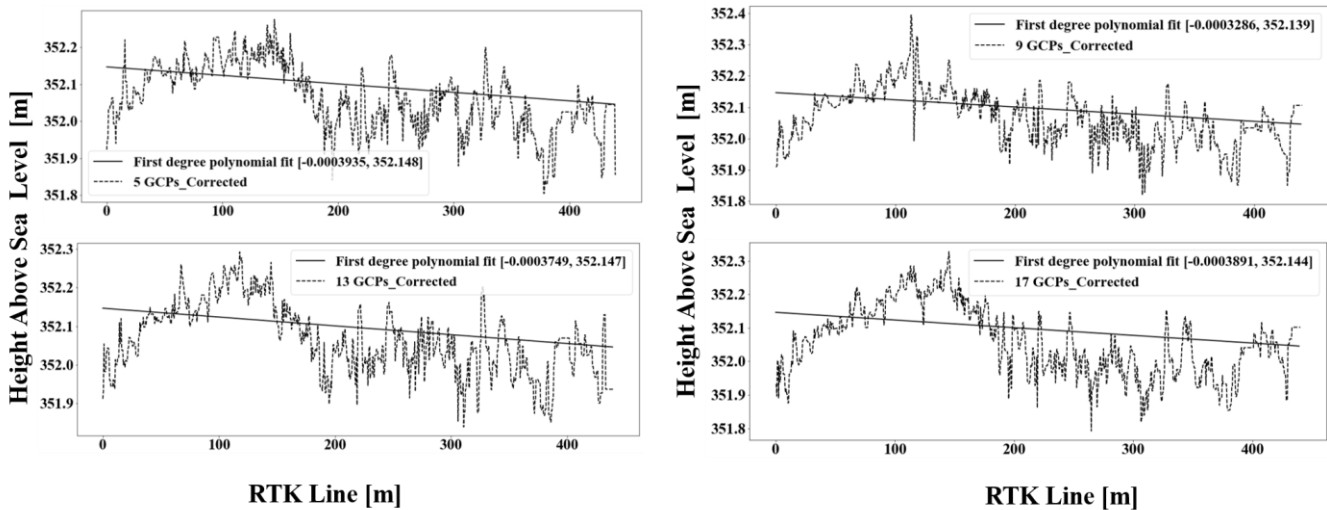

**Fig.B3. First order polynomials through extracted tracks**

Figure B4 shows the relationship between depth and area, as well as the relationship between depth and conveyance. The top
left graph represents the relationship between depth and area at the cross section on the northern part of the terrain. The top
right graph represents the relationship between depth and conveyance at the cross section on the northern part of the terrain.
The two bottom left graphs represents the relationship between depth and area at the cross section on the northern part of the
terrain. The bottom right graph represents the relationship between depth and conveyance at the cross section on the northern
part of the terrain.

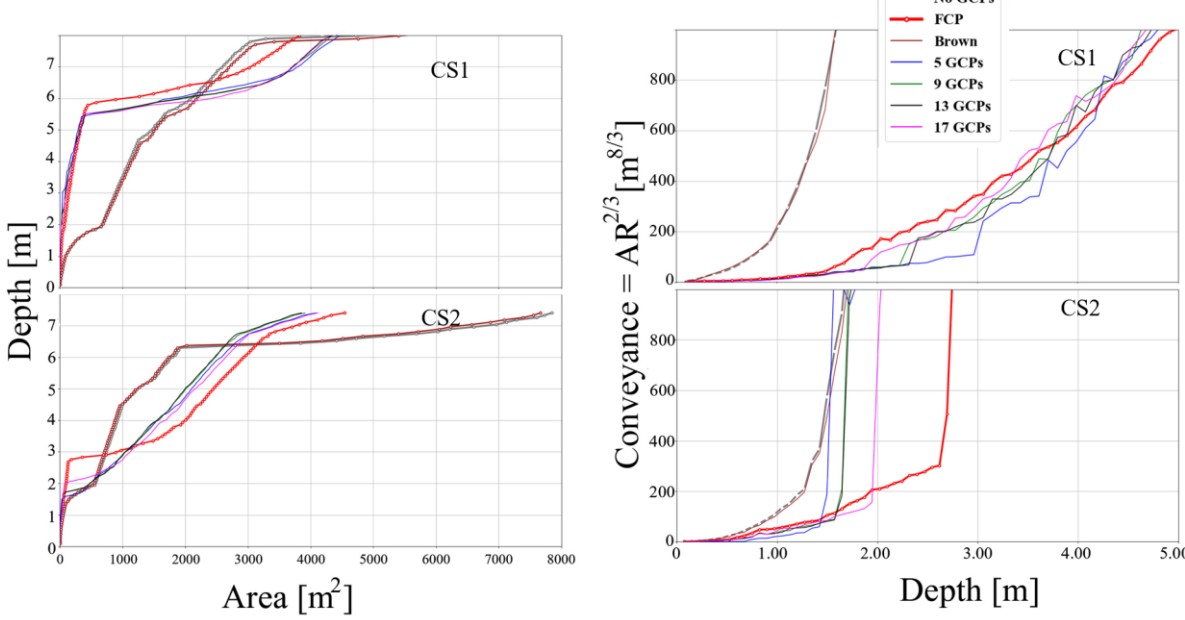

**Fig. B4. Depth vs Area Map and Conveyance vs Depth**

Figure B5 shows the relationships between floodplain width and height above mean sea level, as well as the relationships between wetted perimeter and area. The top left graph represents the relationship between floodplain width and height above mean sea level at the cross section on the northern part of the terrain (CS1). The top right graph represents the relationship between wetted perimeter and area at the cross section on the northern part of the terrain. The bottom left graph represents the relationship between floodplain width and height above mean sea level at the cross section on the northern part of the terrain. The bottom right graph represents the relationship between wetted perimeter and area at the cross section on the northern part of the terrain.

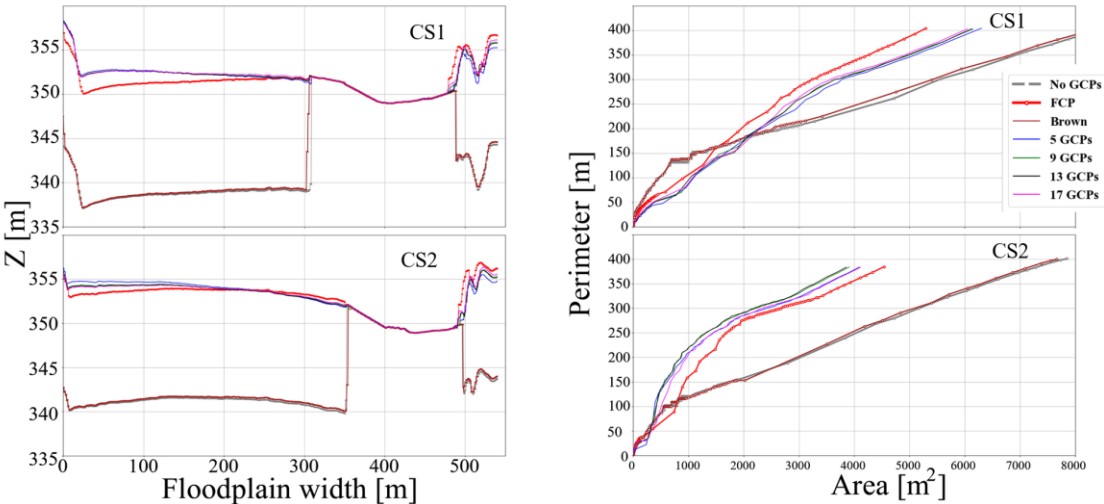


<p style="text-align:center">Fig. B5. Height vs width graph and Perimeter vs Area</p>

## 5 Data Availability

Images used to carry out the ODM photogrammetry can be found on https://doi.org/10.4121/14865225.

## 6 Author Contributions

Hubert Samboko performed conceptualisation, data curation, formal analysis, investigation and writing the original draft. Hessel Winsemius performed conceptualisation, reviewing, editing and supervision. Sten Schurer performed data collection, curation and investigation. Hodson Makurira performed supervision, reviewing and editing. Kawawa Banda performed reviewing and editing. Hubert Savenije performed funds acquisition, supervision, reviewing and editing.

## 7 Competing interests

The authors declare that they have no conflict of interest

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

## 9    Acknowledgments

This work is part of the research programme ZAMSECUR with project number W 07.303.102, which is financed by the Netherlands Organisation for Scientific Research (NWO). This research received and continues to receive support from the University of Zambia and the Zambian Water Resources Management Authority.