# Peer review of "Evaluating low cost topographic surveys for computations of conveyance."

_Geoscientific Instrumentation, Methods and Data Systems, 2021_

## Author Response (AR1)

Response to comments

We acknowledge and appreciate the comments provided by RC1. We find that the comments were accurate in pointing out weaknesses of the manuscript especially with respect to missing descriptions and literature review of key methodologies/outputs. We therefore kindly submit the following responses to each comment made in order to improve the document.

| No | Comments from Reviewer RC1 | Authors Response | Authors Changes |
|---|---|---|---|
| 1 | Title: The title provided is too generic. I suggest to change it focusing on the UAS technologies for surveying. | We note and agree with the comment. The study indeed relies on the results of the UAV products and RTK GNSS (i.e. low cost technologies) | We changed the title to 'EVALUATING LOW COST TOPOGRAPHIC SURVEYS FOR HYDRAULIC RATING' |
| 2 | Abstract. The abstract is too long. It should be a very brief summary of your paper. | We note the lengthiness of the abstract. We identify that we include a long methodology explanation which may not be necessary for an abstract. | Abstract was shortened. This will be achieved by excluding the elaborate methodology explanation |
| 3.1 | The introduction is too generic and doesn't focus on the main questions of the paper. Some references are missed or included only in the following sections. | We identify the shortcomings of the Introduction. Mainly through the specific exclusion of :
 1. Information on the RTK GNSS equipment (Accuracy, cost, method)
 2. The gridding or merging approach of the bathymetry
 3. The doming effect and lens distortion (brown Conrady, Fixed camera parameter FCP etc.)
 4. The influence/importance of the slope | We have significantly altered the introduction to add all the important missing information.
 We changed the introduction such that the novel attributes of the manuscript as well as the objectives are fully described. |
| 3.2 | Some main research questions (e.g. the impact of lens distortion on geometry accuracy) are introduced only in the final sections. | This is a critical missing component. | We added the missing information earlier on in the manuscript. **E.g**. the impact of lens distortion accuracy the observation of slope and the merging of dry and wet river profile |
| 3.3 | Authors should include a complete overview of previous research on this topic (at line 108-111) in order to evidence the added value of their outcomes. | It is noted that there are references necessary to support the claim that **'limited studies have investigated how critical factors** (GCP number and distribution, lens distortion, slope, free and open source software) **can be adjusted to improve hydraulic modelling'** | We added all the missing reference literature on how previous researchers have not focused on 1. How factors can be adjusted to improve elevation models
 2. how to combine the wet and dry bathymetry |
| 4.1 | Please move the research questions at the end of section 1 "Introduction". | Noted | The research questions were be moved from section 2 to the end of section 1 as advised. |
| 4.2 | I suggest to create a new section for "Study site". To this regard, an overview of the morphological and hydraulic characteristics of the river reach can be useful. | To give the reader some insight into the environment within which the study is conducted, we agree with the comment and add a 'study site' section to describe the river characteristics | We described the '**Morphological and hydraulic characteristics of the river** reach' under the study site section |
| 4.3 | Respect to the third research question (line 128), please clarify that the objectives are referred to the error estimation of some variables useful for the indirect estimation of the | The objective was indeed misleading to reader as it may have been misconstrued as estimation of discharge rather than an estimation of proxy | Objective was changed to be more specific
 'What impact does utilising elevation models, reconstructed based on |

| | | | |
|---|---|---|---|
| | discharge (hydraulic conveyance and slope). | variables such a conveyance and slope | different GCP numbers have on conveyance and hydraulic slope' |
| 5 | Sub-section 2.2.1. Line 145. All the configurations used for the study it should be specified at this point. | The various configurations were indeed missing. Especially those to do with the GCP numbers and distributions | We added a paragraph which describes the configurations for the various experiments. Particularly the GCP distribution and density. To aid with configuration visualization, we added an image of the flight pattern which was used to collect the images. |
| 6 | Section 2.2.4. Line 185: The information relative to computer performance can be useful if a comparison between the two software on computation time is achieved. | To be able to authoritatively compare software it is indeed noted that all computer hardware variables need to be mentioned. | We added more information on the computer performance. In addition we added a reference to the minimum hardware requirements according to the software developers |
| 7 | Section 2.3.2. Line 230. I suggest to specify how the GCP points are spatially distributed along the river pattern especially respect to the vertical variability. | Seeing as literature suggests that GCP distribution is critical to achieve good accuracy, it is noted that it would be useful to explain the distribution methodology used. | We describe how the GCP are distributes (E.g. the 2-1-2 or checkerboard formation) and also outline how the maximum and minimum elevations were taken into consideration in order to not only have a representative horizontal distribution, but a vertical distribution as well. |
| 8 | Sub-section 3.2: The configuration Brown-Conrady is not described in the main text. | Brown Conrady is linked to one of the key factors which determine if photogrammetric geometry will be accurate. Its omission in the main text is an error | We added a literature review on the Brown Conrady as well as other calibration models to the main text (Introduction). We will also describe some of the configuration which can deal with the doming effect such as FCP in the main text. |
| 9 | Section 3. A separate discussion section should be added in the main structure of the manuscript. This section should include a comparison with other research studies and should be extended to other aspect that play a role in this analysis, i.e. the flight mission planning and the camera settings. | There are indeed some factors which are not looked at in this topic but, however need to be mentioned. | We added a section which takes a closer look at how other studies have assessed the impact of other factors which are not the subject of this particular manuscript such as flight mission planning, camera setting , flight height, speed, direction, light conditions etc. |
| 10 | Section 3.3. Line 325. It is not clear how the slope is calculated based on photogrammetry products. | An explanation of how slope is derived is missing. | We added a description of the method of slope calculation. This will include a brief explanation of a python module called 'rasterio' which is able to interpret raster images, and therefore extract elevation values (Z) which correspond to the RTK line (the 'true') slope coordinates (X,Y) |
| 11 | Section 3.3. Line 336. This step require more details for a better explanation of the procedure. | A more detailed explanation is necessary to aid the reader. | We break down the procedure into the different processes (extraction, merging, volumising). We then describe each process individually. E.g. the extraction entails overlaying the wet bathymetry on the DEM and cutting out the shape using a special tool in cloud compare' |
| 12 | Conclusion and recommendations. Line 377. Please clarify this point in the section "Methods and material": the number of points used for reconstructions and those for validation. | A description of the exact number of points used was indeed missing. This is in terms of the GCP vs check-points | We will add a description of all the various GCP configurations which were used to 2.2.1 'Flight Plan' |
| 13 | Figures:

- Please improve the overall quality of the figures. | Noted | We either adjusted or redid some of the figures in terms of the actual image quality, labelling and caption |

| 13.1 | Generally, captions are not very descriptive. Please modify accordingly. | Noted | We adjusted all captions such that they fully describe what can be seen in each image |
|---|---|---|---|
| 13.2 | Some figures are not described in the main text (e.g. Figure 11, Figure 5b). | Noted | We made sure that all all images are fully described. Figure 6 in particular has been explained. A step by step description will be added to section 2.2.4 'processing dry and wet bathymetry' |
| 13.3 | In some figure, useful information is missing: the name of cross-sections (figure 7), the measure units (Figure 11), flow direction. | Noted | We will adjust these and other images appropriately |

We acknowledge and appreciate the comments made by RC2. We are greatly encouraged to hear that the reviewer finds the data collected potentially interesting and of practical use to water managers in developing countries. As advised by RC2, we acknowledge the need to reorganise the paper to focus more on the novel attributes such as the 'gridding' approach, low-cost GNSS based bathymetry, RTK line. We therefore submit the following point by point responses to the reviewer comments.

| | Comments from Reviewers RC2 | Authors Response | Authors Changes |
|---|---|---|---|
| 1 | clarify in the abstract that GNSS data are used to characterize the subaqueous bathymetry, and UAVs are only used to map the dry surfaces. | The manuscript indeed misses out on the opportunity to describe much more about the low-cost GNSS. Which we believe could be revolutionary in terms of access to accurate measurements for researchers with smaller budgets. | We adjusted the abstract to clarify that the GNSS is in fact the key tool for the wet bathymetry reconstruction. We also added a description of the system and its costs to the main text. |
| 2 | The description of the UAV flight path is not clear. Was the UAV flown in one direction back and forth ("lawnmower" style) or in two direction back and forth ("checkerboard" style)? | Given that the flight path is important as mechanism that can be manipulated to reduce the doming effect, it is noted that the specific flight method must be clarified. | We added a description of the flight path as well as a figure to aid with visualisation. |
| 3 | The flowchart in Fig 5 needs to be described better. For instance, what is "MVS"? | The flow-chart which describes the SfM processing of the dry bathymetry was indeed not described. Including terms such as Multi view Stereo (MVS) | We redid the image to include fully described terms rather than acronyms and we detail a step by step description of the flowchart in section 2.2.4 'processing dry and wet bathymetry' |
| 4 | clarify how the slope was extracted – was a plane fit to the DEM? Is the slope computed from the average of dry points? | An explanation of how slope is derived is missing. NB* The slope will be compare to the slope of the RTK line (collected using the rolling cart) | We added a description of the method of slope calculation. This includes a brief explanation of a python module called 'rasterio' which is able to interpret raster images, and therefore extract elevation values (Z) which correspond to the RTK line (the 'true') slope coordinates (X,Y). |
| 5 | I believe this is the first mention of "Fixed Camera Parameter". This needs to be described earlier and in more detail. The method is partly described later (line 345) but that is out of place. | The term FCP is indeed misplaced and is supposed to be described in the introduction in-line with methods which can potentially decrease the impact of lens distortion (doming) | We added a description of FCP to the main text, including references and why it is potentially useful. |

| 6 | Figure 13 makes it appear there is a lateral slope of the water surface. Was there? Can we be sure the RTK system is working properly? | We had the opportunity to do simple pre-experimental tests on the accuracy of the RTK system and its accuracy was working. There was no lateral slope of the water surface. However the extreme left bank of the river was inaccessible due to overgrown vegetation. This implies that a small section which would equalise the water levels on the left and right bank is missing. | We provided an explanation for the apparent lateral slope. Unfortunately selecting a location which would satisfy this condition and many other conditions such as a straight reach, accessible flood plain, etc. was not straightforward |
|---|---|---|---|
| 7 | Figures B4 and B5 do not have legends. | Noted | We added legends to both figure sets. |

---

## Author Response (AR2)

We are extremely grateful for the insightful comments made by the reviewer, which improve the quality of Manuscript. We were particularly encouraged by the reviewer comments, which compelled us to revise the optimal GCP number from 9 to 13 due to certain inconsistencies. The clear and concise questions made correction of the minor revisions highly efficient. Below we give a point-by-point response to each of the comments made by the reviewer.

| Comments | Author comment | Changes |
|---|---|---|
| Consider changing the title - seems like it would make more sense to mention channel conveyance rather than hydraulic rating. Perhaps 'Evaluation of low-cost topographic surveys for computations of river conveyance'? | Noted | Corrected to the proposed title 'Evaluation of low-cost topographic surveys for computations of river conveyance.' |
| clarify what 'these' is referring to | Noted | Clarified to indicate these refers to 'open source software' |
| on hydraulics' should just be 'hydraulics' | Noted | Changed to just 'hydraulics' |
| measurement' should be 'measurements' | Noted | Changed to measurements |
| unclear what is meant by 'time validity of the measurement | The statement 'time validity' was indeed unclear. It was referring to changes which might affect the rating curve after due to factors such as flooding , siltation, bed degradation, channel rerouting etc. | An explanation has been added to line 54 which states that what is meant by time validity is the correctness of the rating curve after a period of time |
| I think 'measurements' should be 'calculations' – you are discussing doing a calculation (not measurement) of discharge her | Noted | Changed measurements to calculations |
| Here are more factors to consider' – for what? | The explanation was missing | Corrected to state that factors to consider are for conveyance measurements |
| I believe this is the first place 'GCP' is used in the paper. This should be defined/introduced earlier. | This was indeed the first mention of GCPs and should be mentioned before | We initiate the use of the term GCP in full in line 77 |
| unnecessary apostrophe in 'points' | Noted | Corrected to 'points' |
| not clear what 'high-water bed' means | The term was indeed confusing | We correct the term to mention floodplain instead of high water wet |
| 'know' should be 'known' | noted | Corrected to known |
| Confused about the comparison of Agisoft and ODM RMSEs – with 9 GCPs, the bootstrap box plot appears to indicate that ODM has significantly larger error than Agisoft – approximately twice the error, with non-overlapping box plots. Yet the authors make a point that the results are comparable. Am I missing something? Having 2x error seems like a significant downside to me. | We acknowledge the confusion that might come from the conclusions drawn from selecting 9 as the optimum GCP number. Our selection of this value was based on 3 questions.
 1. Is there an improvement from the previous GCP RMSE values
 2. Does increasing the GCP value to the next value improve on the RMSE value
 3. How does the RMSE value fair in terms of absolute magnitude.
 In our situation we concluded that | The comment presented by the reviewers compels us to adjust the optimum GCP combination to 13.

 13 GCPs satisfies most of the factors we had based our arguments on. With respect to the 2 factors proposed by the reviewer, (a) 13 GCPs has overlap in the box plot and the error. (b) With respect to having 2X the |

| | 1. There was indeed an improvement from 5 to 9 GCPs with respect to RMSE. 2. There was no improvement from 9 to 13 GCPs 3. According to the box plot, the lowest value of RMSE was noted on 9GCPs

The reviewer however makes two valid arguments 1, there is no overlap in the box plot and 2, the error of ODM is twice that of Agisoft | error we identify that ODM is able to limit the RMSE to less than 0.20 m. This is particularly useful because for the purposes of merging with the wet bathymetry <0.20m is sufficient since the accuracy of the wet bathymetry is generally not as accurate because of interpolation. Line 418 |
|---|---|---|
| The results indicate a decrease in the RMSE as we increase the number of GCPs' – in general yes, but the RMSE increases when going from 5 to 9 GCPs before going down. Given 9 GCPs is presented as the optimal number to use, this seems problematic. | In line with the previous comment on the boxplot, we adjust the optimal number to 13 GCPs so that there is consistency in the arguments presented. | In line 418 we justify the use of 13 GCPs as a correction from the previously mentioned 9 GCPs |
| In Figure 14, there appears to be a substantial topographic artifact/mismatch in the upper part of the study area where the wet and dry bathymetry are merged, with noticeably higher elevations in the 'wet' bathymetry relative to the adjacent 'dry' bathymetry. This isn't addressed in the paper, but seems like it may be an important issue, especially if this were to be used for hydraulic modeling at some point. | The mismatch in elevations on the northern section is noted. It is increasingly evident that there is a need to affirm the importance of accurate wet bathymetric surveys if merging and subsequent accurate hydraulic modeling is to be achieved. | We propose to add this comment into the conclusion and recommendations section. The proposal is to either cut off the section which present significant mismatches in elevations or to increase wet bathymetry transects such that algorithms used to merge the two bathymetries (in this case Cloud Compare) have access to more transects which improve interpolation of the elevations. Line 544 |
| this text on FCP was already presented in the introduction. | This text was indeed unnecessary repetition. | We removed it from the text. |
| it seems that the point of this slope analysis is to say that for these conditions the slope should not be estimated from the SfM data under any circumstances. Suggest saying something explicit like this when the data are presented. | This is noted | We add a statement affirming the inapplicability of SfM for slope derivation in line 509 |
| relatively – relatively what? | Noted, an omission of the word 'quickly' | The word has been added to line 517 |
| seems that the conveyance is more impacted by the quality of the 'wet' bathymetry collected by the GNSS than the 'dry' SfM bathymetry. This point could be emphasized a bit more when discussing the results. | This is noted and is interestingly related to the previously mentioned comment on figure 14 were a mismatch in elevation could affect the quality of results for both conveyance and discharge modelling. We are however encouraged to note that our study moves towards accurate reconstruction of not only the dry bathymetry but the wet bathymetry as well. | We add a statement in line 540 to emphasises the importance of the wet bathymetry |